# SQ Lower Bounds for Learning Mixtures of Linear Classifiers

**Ilias Diakonikolas**
University of Wisconsin-Madison
ilias@cs.wisc.edu

**Daniel M. Kane**
University of California, San Diego
dakane@cs.ucsd.edu

**Yuxin Sun**
University of Wisconsin-Madison
yxsun@cs.wisc.edu

## Abstract

We study the problem of learning mixtures of linear classifiers under Gaussian covariates. Given sample access to a mixture of $r$ distributions on $\mathbb{R}^n$ of the form $(\mathbf{x}, y_\ell)$, $\ell \in [r]$, where $\mathbf{x} \sim \mathcal{N}(0, \mathbf{I}_n)$ and $y = \mathrm{sign}(\langle \mathbf{v}_\ell, \mathbf{x} \rangle)$ for an unknown unit vector $\mathbf{v}_\ell$, the goal is to learn the underlying distribution in total variation distance. Our main result is a Statistical Query (SQ) lower bound suggesting that known algorithms for this problem are essentially best possible, even for the special case of uniform mixtures. In particular, we show that the complexity of any SQ algorithm for the problem is $n^{\mathrm{poly}(1/\Delta)\log(r)}$, where $\Delta$ is a lower bound on the pairwise $\ell_2$-separation between the $\mathbf{v}_\ell$'s. The key technical ingredient underlying our result is a new construction of spherical designs that may be of independent interest.

## 1   Introduction

The motivation behind this work is to understand the computational complexity of learning high-dimensional latent variable (aka mixture) models. The task of learning various mixture models has a long history in statistics with an early focus on sample efficiency, starting with the pioneering work of Karl Pearson [Pea94] on learning Gaussian mixtures. During the past decades, an extensive line of work in machine learning and theoretical computer science has made significant progress on the computational aspects of this general question for a range of mixture models, including mixtures of high-dimensional Gaussians [Das99, AK01, VW02, AM05, KSV08, BV08, MV10, RV17, HL18, DKS18], mixtures of linear regressions [SJA16, LL18, CLS20, DK20], and more generally mixtures of experts [JJ94, XA09, ME14, MVKO19, HYJ22].

In this paper, we focus on mixtures of *linear classifiers*, a classical supervised probabilistic model that has been intensely investigated from both statistical and algorithmic standpoints [SIM14, LLYH17, GMP20, CDV22]. A linear classifier (or halfspace) is any Boolean function $h : \mathbb{R}^n \to \{\pm 1\}$ of the form $h(\mathbf{x}) = \mathrm{sign}(\langle \mathbf{v}, \mathbf{x} \rangle)$, where $\mathbf{v} \in \mathbb{R}^n$ is known as the weight vector and the univariate function $\mathrm{sign}$ is defined by $\mathrm{sign}(t) = 1$ for $t \geq 0$ and $\mathrm{sign}(t) = -1$ otherwise. For an integer $r \geq 2$, we can now formally describe an $r$-mixture of linear classifiers. The parameters of the model contain $r$ unknown positive weights $w_1, \ldots, w_r$ with $\sum_{\ell=1}^{r} w_\ell = 1$, and $r$ unknown unit vectors $\mathbf{v}_1, \ldots, \mathbf{v}_r \in \mathbb{R}^n$. A random sample is drawn from the underlying distribution $D(\mathbf{x}, y)$ as follows: the sample oracle selects the index $\ell \in [r]$ with probability $w_\ell$, and we then receive a random point $(\mathbf{x}, y) \in \mathbb{R}^n \times \{\pm 1\}$ where $\mathbf{x} \sim \mathcal{N}(0, \mathbf{I}_n)$ and $y = \mathrm{sign}(\langle \mathbf{v}_\ell, \mathbf{x} \rangle)$. The goal is to approximately estimate the model either by learning the underlying distribution $D(\mathbf{x}, y)$ in total variation distance (density estimation), or by approximately recovering the hidden parameters, i.e.,

37th Conference on Neural Information Processing Systems (NeurIPS 2023).

$w_\ell$ and $\mathbf{v}_\ell$, $\ell \in [r]$ (parameter estimation). An algorithm for parameter estimation can be used for density estimation (because closeness in parameters can be shown to imply closeness in total variation distance). In that sense, parameter estimation is a harder problem.

Before we proceed to describe prior algorithmic work, we provide basic background on the sample complexity of the problem. We start by noting that density estimation for $r$-mixtures of linear classifiers on $\mathbb{R}^n$ is information-theoretically solvable using $\mathrm{poly}(n, r)$ samples (with the optimal bound being $\tilde{\Theta}(nr/\epsilon^2)$ for total variation error $\epsilon$) without any assumptions on the components. In contrast, for parameter estimation to be information-theoretically solvable with polynomial sample complexity, some further assumptions are needed. The typical assumption involves some kind of pairwise separation between the component vectors and a lower bound on the mixing weights. Let $\Delta \overset{\text{def}}{=} \min_{i \neq j}\{\|\mathbf{v}_i - \mathbf{v}_j\|_2, \|\mathbf{v}_i + \mathbf{v}_j\|_2\} > 0$ be the pairwise separation between the components and $w_{\min}$ be the minimum mixing weight. Under these assumptions, the parameter estimation problem is solvable using $\mathrm{poly}(n, r, 1/\Delta, 1/w_{\min}, 1/\epsilon)$ samples to achieve parameter error of $\epsilon$. In both cases, the term "information-theoretically solvable" is used to mean that a *sample-efficient* algorithm exists, without any constraints on computational efficiency. The main question addressed in this paper is to what extent a *computationally efficient* learning algorithm exists.

On the algorithmic front, [CDV22] gave provable parameter estimation algorithms under a $\Delta$-separation assumption. Specifically, [CDV22] provided two algorithms with different complexity guarantees. Their first algorithm has sample and computational complexity $\mathrm{poly}(n^{O(\log(r)/\Delta^2)}, 1/w_{\min}, 1/\epsilon)$, while their second algorithm has complexity $\mathrm{poly}((n/\Delta)^r, 1/w_{\min}, 1/\epsilon)$. Here we focus on settings where the number of components $r$ is large and cannot be viewed as a constant. For the sake of intuition, it is instructive to simplify these upper bounds for the regime of uniform mixtures (corresponding to the special case that $w_\ell = 1/r$ for all $\ell \in [r]$) and $\epsilon$ is not too small. For this regime, the complexity upper bound achieved in [CDV22] is $\min\{n^{O(\log(r)/\Delta^2)}, (n/\Delta)^{O(r)}\}$. Concretely, if the separation $\Delta$ is $\Omega(1)$ or even $1/\mathrm{polylog}(r)$, the first term yields a quasi-polynomial upper bound. On the other hand, for $\Delta = O(1/r^c)$, for a constant $c > 0$, the resulting upper bound is $n^{\mathrm{poly}(r)}$. In both regimes, we observe a *super-polynomial* gap between the information-theoretic sample complexity — which is $\mathrm{poly}(n, r, \Delta)$ — and the sample complexity of the [CDV22] algorithms. It is thus natural to ask if this gap is inherent.

> *What is the complexity of learning mixtures of linear classifiers?*
> *Is there an algorithm with significantly better* sample-time tradeoff*?*

We study these questions in a well-studied restricted model of computation, known as the Statistical Query (SQ) model [Kea98] (and, via [BBH+20], also for low-degree polynomial tests). *Our main result is that in both of these models the complexity of the above-mentioned algorithms is essentially best possible.* Along the way, we establish new results on the existence of spherical designs on the unit sphere that may be of independent interest.

## 1.1 Our Results

**Basics on SQ Model** SQ algorithms are a class of algorithms that, instead of having direct access to samples, are allowed to query expectations of bounded functions of the distribution. Formally, an SQ algorithm has access to the following standard oracle.

**Definition 1.1** (STAT Oracle). Let $D$ be a distribution on $\mathbb{R}^n$. A *Statistical Query* is a bounded function $f : \mathbb{R}^n \to [-1, 1]$. For $\tau > 0$, the $\mathrm{STAT}(\tau)$ oracle responds to the query $f$ with a value $v$ such that $|v - \mathbf{E}_{\mathbf{x} \sim D}[f(\mathbf{x})]| \leq \tau$. A *Statistical Query (SQ) algorithm* is an algorithm whose objective is to learn an unknown distribution $D$ by making adaptive calls to the $\mathrm{STAT}(\tau)$ oracle.

The SQ model was introduced in [Kea98]. Subsequently, the model has been extensively studied in a range of contexts [Fel16]). The class of SQ algorithms is broad and captures a range of known supervised learning algorithms. More broadly, several known algorithmic techniques in machine learning are known to be implementable using SQs [FGR+17, FGV17].

Our main result is a near-optimal SQ lower bound for mixtures of linear classifiers that applies even for the uniform case. Specifically, we establish the following:

**Theorem 1.2** (Main Result: SQ Lower Bound for Uniform Mixtures)**.** *Let $\epsilon \leq c\Delta/r$ for some universal constant $c > 0$ sufficiently small. Then, any SQ algorithm that learns a uniform mixture of*

*linear classifiers with directions $\mathbf{v}_1, \ldots, \mathbf{v}_r \in \mathbb{S}^{n-1}$ satisfying $\min_{i \neq j}\{\|\mathbf{v}_i - \mathbf{v}_j\|_2, \|\mathbf{v}_i + \mathbf{v}_j\|_2\} \geq \Omega(\Delta)$ for some $r^{-1/10} \leq \Delta < 1$, within error $\epsilon$ in total variation distance must either use queries of tolerance $n^{-\mathrm{poly}(1/\Delta)\log r}$, or make at least $2^{n^{\Omega(1)}}$ queries.*

Informally speaking, Theorem 1.2 shows that no SQ algorithm can perform density estimation for uniform mixtures of linear classifiers to small accuracy with a sub-exponential in $n^{\Omega(1)}$ many queries, unless using queries of very small tolerance – that would require at least $n^{\mathrm{poly}(1/\Delta)\log r}$ samples to simulate. This result can be viewed as a near-optimal information-computation tradeoff for the problem, within the class of SQ algorithms. In more detail, for $\Delta = \Omega(1)$, we obtain a quasi-polynomial SQ lower bound of $n^{\Omega(\log r)}$; while for $\Delta = 1/r^c$, for some constant $0 < c < 1/2$, we obtain an SQ lower bound of $n^{\mathrm{poly}(r)}$. In both cases, our SQ lower bounds qualitatively match the previously known algorithmic guarantees [CDV22] (that are easily implementable in the SQ model).

A conceptual implication of Theorem 1.2 is that the uniform (i.e., equal mixing weights) case is essentially as hard as the general case for density estimation of these mixtures. In contrast, for related mixture models, specifically for mixtures of Gaussians, there is recent evidence that restricting the weights may make the problem computationally easier [BS21].

**Remark 1.3.** We note that the condition $\Delta \geq r^{-c}$, for some constant $0 < c < 1$, is necessary in the statement of Theorem 1.2 for the following reason: the algorithmic result of [CDV22] has sample and computational complexity $\min\{n^{O(\log(r)/\Delta^2)}, (n/\Delta)^{O(r)}\}$, which will be $(n/\Delta)^{O(r)} \ll n^{\mathrm{poly}(1/\Delta)\log r}$ if $\Delta$ is sufficiently small by inverse polynomial in $r$.

**Remark 1.4.** Our SQ lower bound result has immediate implications to another well-studied restricted computational model — that of low-degree polynomial tests [HS17, HKP$^+$17, Hop18]. [BBH$^+$20] established that (under certain assumptions) an SQ lower bound also implies a qualitatively similar lower bound in the low-degree model. This connection can be used as a black-box to show a similar lower bound for low-degree polynomials.

The key technical ingredient required for our SQ lower bound is a theorem establishing the existence of spherical designs with appropriate properties. The definition of a spherical design follows.

**Definition 1.5** (Spherical Design). Let $t$ be an odd integer. A set of points $\mathbf{x}_1, \ldots, \mathbf{x}_r \in \mathbb{S}^{n-1}$ is called a spherical $t$-design if $\mathbf{E}[p(\mathbf{x})] = \frac{1}{r}\sum_{i=1}^r p(\mathbf{x}_i)$ holds for every homogeneous $n$-variate polynomial $p$ of degree $t$, where the expectation is taken over the uniform distribution on the unit sphere $\mathbb{S}^{n-1}$.

We note that this definition differs slightly from the usual definition of spherical design, which requires that the equation in Definition 1.5 holds for all polynomials $p$ of degree *at most* $t$. However, by multiplying by powers of $\|\mathbf{x}\|_2^2$, we note that our definition implies that the equation holds for every odd polynomial of degree at most $t$, and it is sufficient for the requirement of establishing our SQ lower bound.

Spherical designs have been extensively studied in combinatorial design theory and a number of efficient constructions are known, see, e.g., [Ban79, DGS91, GP11, BRV13, Kan15, Wom18]. However, none of the known constructions seem to be compatible with our separation assumptions. We establish the following result that may be of independent interest in this branch of mathematics.

**Theorem 1.6** (Efficient Spherical Design). *Let $t$ be an odd integer and $r \geq \binom{n+2t-1}{n-1}^5$. Let $\mathbf{y}_1, \ldots, \mathbf{y}_r$ be uniform random vectors over $\mathbb{S}^{n-1}$. Then, with probability at least 99/100, there exist unit vectors $\mathbf{z}_i \in \mathbb{S}^{n-1}$ very close to $\mathbf{y}_i$ such that $(\mathbf{z}_1, \ldots, \mathbf{z}_r)$ form a spherical $t$-design.*

We mention here that the optimal sample complexity for the existence of spherical $t$-design (for even $t$) is $r = \Theta\left(\binom{n+t}{n}\right)$ and our result matches this within a polynomial factor. Theorem 1.6 is essential for our construction of moment-matching. Roughly speaking, the resulting mixture matches moments in the sense we require if and only if the weight vectors $(\mathbf{v}_1, \ldots, \mathbf{v}_r)$ form a *spherical design*. We mention that the closeness between $\mathbf{z}_i$ and $\mathbf{y}_i$ is necessary since this leads to the separation of the hidden weight vectors in the mixture of linear classifiers. In order to guarantee the pairwise separation of the weight vectors $\mathbf{v}_i$'s, we note that with high probability the $\mathbf{y}_i$'s are pairwise separated and therefore if $\mathbf{z}_i$ is sufficiently close to $\mathbf{y}_i$ they will be too.

## 1.2 Technical Overview

Our starting point is the SQ lower bound technique of [DKS17] and its generalization in [DKPZ21]. In particular, our overall strategy is to find a mixture of homogeneous halfspaces in some "small" number $m$ of dimensions that matches its first $k$ moments with the distribution on $(\mathbf{x}, y)$, where $y$ is independent of $\mathbf{x}$. By appropriately embedding this low-dimensional construction into $\mathbb{R}^n$ via a random projection of $\mathbb{R}^n \to \mathbb{R}^m$, the SQ lower bound machinery of [DKPZ21] can be shown to imply that learning the projection requires either $2^{n^{\Omega(1)}}$ queries or some query of accuracy $n^{-\Omega(k)}$. This generic step reduces our problem to finding an appropriate $m$-dimensional construction. Such a construction is the main technical contribution of this work.

Note that specifying a mixture of halfspaces is equivalent to specifying a probability distribution with support consisting of at most $r$ unit vectors and specifying the orthogonal vectors of the halfspaces. It is not hard to see that the resulting mixture matches moments in the sense we require if and only if these vectors form a *spherical k-design* [DGS91]: namely, that for any odd polynomial $p$ of degree less than $k$ that the average value of $p$ over our distribution is the same as the average value of $p$ over the unit sphere (namely, equal to zero). The bulk of our problem now reduces to finding constructions of weighted designs, where: (1) The support size of the design is relatively small. (2) The points in the design are pairwise separated. (3) For equally weighted mixtures, we require that the weight of each vector in the design is uniform.

In $m = 2$ two dimensions, there is a relatively simple explicit construction that can be given. In particular, if we take $k$ evenly spaced points over the unit circle for some odd $k$, this matches the first $k-1$ moments and has separation $\Omega(1/k)$. (This is similar to an SQ construction in [DKKZ20] for a different setting.) Unfortunately, we cannot obtain better separation in two dimensions, since any $k$ unit vectors in two dimensions will necessarily have some pair separated by at most $O(1/k)$. Therefore, if we want constructions with greater separation, we will need to pick larger values of $m$. Indeed, we show (Proposition 3.2) that a *random* collection of approximately $m^{O(k)}$ points can form the support of an appropriate design. This can be proved by applying linear programming duality. Unfortunately, this argument does not allow us to control the mixing weights. Specifically, it may be the case that the minimum weight is exponentially small which seems unnatural in practice.

The case of equal weights requires a significantly more sophisticated argument. In this case, we need to find a spherical design with uniform weight. Indeed, merely selecting a random support will no longer work. Although there is an extensive literature on finding efficient spherical designs [Ban79, DGS91, GP11, BRV13, Kan15, Wom18], none of the known constructions seem to be compatible with our separation assumptions. In particular, [BRV13] proves that for each $r \geq c_m k^m$, there exists a spherical $k$-design, where $c_m$ is a constant depending only on the dimension $m$. Although it is plausible that their construction can be adapted to satisfy our separation requirement, their sample complexity has a potentially bad dependence on the dimension $m$. On the other hand, [Kan15] achieves the optimal sample complexity $r$ up to polynomials for all $m$ and $k$. However, the construction in [Kan15] definitely does not satisfy our separation requirement. We note that the sample complexity of $r = \Theta\left(\binom{m+k}{k}\right)$ should be optimal, and our results are polynomial in this bound. In conclusion, to establish our SQ lower bound, we need to prove a new technical result. Our starting point to that end will be the work of [BRV13]. The basic plan here will be to select random unit vectors $\mathbf{y}_1, \mathbf{y}_2, \ldots, \mathbf{y}_r \in \mathbb{R}^n$, and then — applying topological techniques — to show that there is some small perturbation set $\mathbf{x}_1, \mathbf{x}_2, \ldots, \mathbf{x}_r \in \mathbb{R}^n$ that is a spherical design. In particular, we will find a continuous function $F$ mapping degree-$k$ odd polynomials to sets of $r$ unit vectors such that, for all unit-norm polynomials $p$, if $F(p) = (\mathbf{z}_1, \ldots, \mathbf{z}_r)$ then $\sum_{i=1}^{r} p(\mathbf{z}_i) > 0$. Given this statement, a standard fixed point theorem [CC06] will imply that our design can be found as $F(q)$ for some $q$ with norm less than one. To construct the mapping $F$, we start with the points $\mathbf{y}_1, \ldots, \mathbf{y}_r$ and perturb each $\mathbf{y}_i$ in the direction of $\nabla_o p(\mathbf{y}_i)$ in order to try to increase the average value of $p$, where $\nabla_o p(\mathbf{y}_i)$ is the component of $\nabla p(\mathbf{y}_i)$ orthogonal to the direction $\mathbf{y}_i$. Intuitively, this construction should work because with high probability the average value of $p(\mathbf{y}_i)$ is already small (since the empirical average should approximate the true average), the average gradient of $p(\mathbf{y}_i)$ is not too small, and the contributions to $p(\mathbf{z}_i)$ coming from higher-order terms will not be large as long as $\mathbf{z}_i$ is sufficiently close to $\mathbf{y}_i$. These facts can all be made precise with some careful analysis of spherical harmonics (Lemma 4.5).

Finally, we need to show that the hardness of learning the appropriate projection implies hardness of learning the mixture. By standard facts, it suffices to show that mixtures of linear classifiers

are far from the distribution $(\mathbf{x}, y)$, where the uniform label $y$ is independent of $\mathbf{x}$, in total variation distance. To show this, we prove that the total variation distance between any projection and the distribution where $y$ is independent of $\mathbf{x}$ is at least $\Omega(\Delta/r)$ (Lemma 3.8).

## 2 Preliminaries

For $n \in \mathbb{Z}_+$, we denote $[n] \overset{\text{def}}{=} \{1, \ldots, n\}$. For two distributions $p, q$ over a probability space $\Omega$, let $d_{\mathrm{TV}}(p, q) = \sup_{S \subseteq \Omega} |p(S) - q(S)|$ denote the total variation distance between $p$ and $q$. In this article, we typically use small letters to denote random variables and vectors. For a real random variable $x$, we use $\mathbf{E}[x]$ to denote the expectation. We use $\mathbf{Pr}[\mathcal{E}]$ and $\mathbb{I}[\mathcal{E}]$ for the probability and the indicator of event $\mathcal{E}$. Let $\mathcal{N}_n$ denote the standard $n$-dimensional Gaussian distribution and $\mathcal{N}$ denote the standard univariate Gaussian distribution. Let $\mathbb{S}^{n-1} = \{\mathbf{x} \in \mathbb{R}^n : \|\mathbf{x}\|_2 = 1\}$ denote the $n$-dimensional unit sphere. For a subset $S \subseteq \mathbb{R}^n$, we will use $\mathcal{U}(S)$ to denote the uniform distribution over $S$. We will use small boldface letters for vectors and capital boldface letters for matrices. Let $\|\mathbf{x}\|_2$ be the $\ell^2$-norm of the vector $\mathbf{x} \in \mathbb{R}^n$. For vectors $\mathbf{u}, \mathbf{v} \in \mathbb{R}^n$, we use $\langle \mathbf{u}, \mathbf{v} \rangle$ to denote their inner product. We denote by $\mathcal{L}^2(\mathbb{R}^n, \mathcal{N}_n)$ the function space of all functions $f : \mathbb{R}^n \to \mathbb{R}$ such that $\mathbf{E}_{\mathbf{z} \in \mathcal{N}_n}[f^2(\mathbf{z})] < \infty$. The usual inner product for this space is $\mathbf{E}_{\mathbf{z} \in \mathcal{N}_n}[f(\mathbf{z})g(\mathbf{z})]$.

For a matrix $\mathbf{P} \in \mathbb{R}^{m \times n}$, we denote $\|\mathbf{P}\|_2, \|\mathbf{P}\|_F$ to be its spectral norm and Frobenius norm respectively. For a tensor $A \in (\mathbb{R}^n)^{\otimes k}$, let $\|A\|_2$ denote its spectral norm and $\|A\|_F$ denote its Frobenius norm. We will use $A_{i_1, \ldots, i_k}$ to denote the coordinate of the $k$-tensor $A$ indexed by the $k$-tuple $(i_1, \ldots, i_k)$. The inner product between $k$-tensors is defined by thinking of tensors as vectors with $n^k$ coordinates. We use the framework of Statistical Query (SQ) algorithms for problems over distributions [FGR+17] and require the following standard definition.

**Definition 2.1** (Decision/Testing Problem over Distributions). Let $D$ be a distribution and $\mathcal{D}$ be a family of distributions over $\mathbb{R}^n$. We denote by $\mathcal{B}(\mathcal{D}, D)$ the decision (or hypothesis testing) problem in which the input distribution $D'$ is promised to satisfy either (a) $D' = D$ or (b) $D' \in \mathcal{D}$, and the goal of the algorithm is to distinguish between these two cases.

**Basics on VC-Inequality** We recall the definition of VC dimension for a set system.

**Definition 2.2** (VC-Dimension). For a class $\mathcal{C}$ of boolean functions on a set $\mathcal{X}$, the *VC-dimension* of $\mathcal{C}$ is the largest integer $d$ such that there exist $d$ points $x_1, \ldots, x_d \in \mathcal{X}$ such that for any boolean function $g : \{x_1, \ldots, x_d\} \to \{\pm 1\}$, there exists an $f \in \mathcal{C}$ satisfying $f(x_i) = g(x_i), 1 \leq i \leq d$.

We will use the following probabilistic inequality.

**Lemma 2.3** (VC inequality, see, e.g., [Ver18]). *Let $\mathcal{C}$ be a class of boolean functions on $\mathcal{X}$ with VC-dimension $d$, and let $X$ be a distribution on $\mathcal{X}$. Let $\epsilon > 0$ and let $N$ be an integer at least a sufficiently large constant multiple of $d/\epsilon^2$. Then, if $X_1, X_2, \ldots, X_N$ are i.i.d. samples from $X$, we have that:*

$$\mathbf{Pr}\left[\sup_{f \in \mathcal{C}} \left| \frac{\sum_{j=1}^{N} f(X_j)}{N} - \mathbf{E}[f(X)] \right| > \epsilon \right] = \exp(-\Omega(N\epsilon^2)).$$

To apply the VC inequality in our context, we additionally need the following fact which gives the VC-dimension of the class of bounded-degree polynomial threshold functions (PTFs).

**Fact 2.4.** *Let $\mathcal{C}_k$ denote the class of degree-$k$ polynomial threshold functions (PTFs) on $\mathbb{R}^m$, namely the collection of functions of the form $f(\mathbf{x}) = \mathrm{sign}(p(\mathbf{x}))$ for some degree at most $k$ real polynomial $p$. Then, the VC-dimension of $\mathcal{C}_k$ is $\mathrm{VC}(\mathcal{C}_k) = \binom{m+k}{k}$.*

## 3 Warmup: SQ Lower Bounds for General Weight Mixtures

**SQ Lower Bound Machinery.** We start by defining the family of distributions that we will use to prove our SQ hardness result.

**Definition 3.1** ([DKPZ21]). Given a function $g : \mathbb{R}^m \to [-1, +1]$, we define $\mathcal{D}_g$ to be the class of distributions over $\mathbb{R}^n \times \{\pm 1\}$ of the form $(\mathbf{x}, y)$ such that $\mathbf{x} \sim \mathcal{N}_n$ and $\mathbf{E}[y \mid \mathbf{x} = \mathbf{z}] = g(\mathbf{U}\mathbf{z})$, where $\mathbf{U} \in \mathbb{R}^{m \times n}$ with $\mathbf{U}\mathbf{U}^{\intercal} = \mathbf{I}_m$.

The following proposition states that if $g$ has zero low-degree moments, then distinguishing $\mathcal{D}_g$ from the distribution $(\mathbf{x}, y)$ with $\mathbf{x} \sim \mathcal{N}_n, y \sim \mathcal{U}(\{\pm 1\})$ is hard in the SQ model.

**Proposition 3.2** ([DKPZ21])**.** *Let* $g : \mathbb{R}^m \to [-1, 1]$ *be such that* $\mathbf{E}_{\mathbf{x} \sim \mathcal{N}_m}[g(\mathbf{x})p(\mathbf{x})] = 0$, *for every polynomial* $p : \mathbb{R}^m \to \mathbb{R}$ *of degree less than* $k$, *and* $\mathcal{D}_g$ *be the class of distributions from Definition 3.1. Then, if* $m \le n^a$, *for some constant* $a < 1/2$, *any SQ algorithm that solves the decision problem* $\mathcal{B}(\mathcal{D}_g, \mathcal{N}_n \times \mathcal{U}(\{\pm 1\}))$ *must either use queries of tolerance* $n^{-\Omega(k)}$, *or make at least* $2^{n^{\Omega(1)}}$ *queries.*

We will apply Proposition 3.2 to establish our SQ lower bound for learning mixtures of linear classifiers. The main technical contribution required to achieve this is the construction of a class of distributions $\mathcal{D}_g$, where each element in $\mathcal{D}_g$ represents a distribution of mixture of linear classifiers. In particular, we will carefully choose some appropriate unit vectors $\mathbf{v}_1, \dots, \mathbf{v}_r \in \mathbb{R}^m$ and non-negative weights $w_1, \dots, w_r$ with $\sum_{\ell=1}^r w_\ell = 1$, such that $\mathbf{v}_\ell$ are pairwise separated by $\Delta > 0$.

Let $g(\mathbf{z}) = \sum_{\ell=1}^r w_\ell \text{sign}(\mathbf{v}_\ell^\intercal \mathbf{z}), \mathbf{z} \in \mathbb{R}^m$. For an arbitrary matrix $\mathbf{U} \in \mathbb{R}^{m \times n}$ with $\mathbf{U}\mathbf{U}^\intercal = \mathbf{I}_m$, we denote by $D_{\mathbf{U}}$ the instance of mixture of linear classifiers with weight vectors $\mathbf{U}^\intercal \mathbf{v}_1, \dots, \mathbf{U}^\intercal \mathbf{v}_r$ and weights $w_1, \dots, w_r$. In this way, we have that

$$\mathbf{E}_{(\mathbf{x}, y) \sim D_{\mathbf{U}}}[y \mid \mathbf{x} = \mathbf{z}] = \sum_{\ell=1}^r w_\ell \text{sign}((\mathbf{U}^\intercal \mathbf{v}_\ell)^\intercal \mathbf{z}) = \sum_{\ell=1}^r w_\ell \text{sign}(\mathbf{v}_\ell^\intercal \mathbf{U}\mathbf{z}) = g(\mathbf{U}\mathbf{z}), \mathbf{z} \in \mathbb{R}^n.$$

## 3.1 Low-degree Moment Matching

The following proposition shows that there exist unit vectors $\mathbf{v}_1, \dots, \mathbf{v}_r$ and non-negative weights $w_1, \dots, w_r$ with $\sum_{\ell=1}^r w_\ell = 1$ such that $\mathbf{v}_\ell$ are pairwise separated by some parameter $\Delta > 0$ and the low-degree moments of $g$ vanish. Note that since $g$ is odd (as the sign function is odd), we only require $\mathbf{E}_{\mathbf{z} \in \mathcal{N}_m}[g(\mathbf{z})p(\mathbf{z})] = 0$ for every odd polynomial $p$ of degree less than $k$.

**Proposition 3.3.** *Let* $m = \frac{c \log r}{\log(1/\Delta)}$, *where* $r^{-1/10} \le \Delta < 1$ *and* $1.99 \le c \le 2$ *is a universal constant. Let* $k$ *be a positive integer such that* $r \ge C \binom{m+k}{k}$ *for some constant* $C > 0$ *sufficiently large. There exist vectors* $\mathbf{v}_1, \dots, \mathbf{v}_r$ *over the unit sphere* $\mathbb{S}^{m-1}$ *and non-negative weights* $w_1, \dots, w_r$ *with* $\sum_{\ell=1}^r w_\ell = 1$, *such that*

- $\mathbf{E}_{\mathbf{z} \sim \mathcal{N}_m}[g(\mathbf{z})p(\mathbf{z})] = 0$ *holds for every odd polynomial* $p$ *of degree less than* $k$, *where* $g(\mathbf{z}) = \sum_{\ell=1}^r w_\ell \text{sign}(\mathbf{v}_\ell^\intercal \mathbf{z})$.

- $\|\mathbf{v}_i + \mathbf{v}_j\|_2, \|\mathbf{v}_i - \mathbf{v}_j\|_2 \ge \Omega(\Delta), \forall 1 \le i < j \le r$.

The proof proceeds as follows: We uniformly sample unit vectors $\mathbf{v}_1, \dots, \mathbf{v}_r \in \mathbb{S}^{m-1}$. We will prove by the probabilistic method that both statements in Proposition 3.3 hold with high probability. We begin by showing that $\mathbf{Pr}[\min_{1 \le i < j \le r}\{\|\mathbf{v}_i - \mathbf{v}_j\|_2, \|\mathbf{v}_i + \mathbf{v}_j\|_2\} < O(\Delta)]$ is small. By symmetry, it suffices to bound $\mathbf{Pr}[\min_{1 \le i < j \le r} \|\mathbf{v}_i - \mathbf{v}_j\|_2 < O(\Delta)]$. Let $\Theta = \min_{1 \le i < j \le r} \arccos\langle \mathbf{v}_i, \mathbf{v}_j \rangle$. We require the following facts:

**Fact 3.4** (Proposition 3.5 in [BRS+18])**.** $\mathbf{Pr}\left[\Theta \ge \gamma r^{-\frac{2}{m-1}}\right] \ge 1 - \frac{\kappa_{m-1}}{2}\gamma^{m-1}$, *where* $\kappa_m = \frac{\Gamma((m+1)/2)}{m\sqrt{\pi}\Gamma(m/2)} \in \left[\frac{1}{m}\sqrt{\frac{m-1}{2\pi}}, \frac{1}{m}\sqrt{\frac{m+1}{2\pi}}\right]$.

Applying Fact 3.4 by taking $\gamma = \left(\frac{1}{50\kappa_{m-1}}\right)^{\frac{1}{m-1}}$ and $r = \frac{1}{\sqrt{50\kappa_{m-1}\Delta^{m-1}}}$ yields that $\mathbf{Pr}[\Theta \ge \Delta] \ge 99/100$. Given some fixed random vectors $\mathbf{v}_1, \dots, \mathbf{v}_r \in \mathbb{S}^{m-1}$, we will prove that with high probability there exist non-negative weights $w_1, \dots, w_r$ with $\sum_{\ell=1}^r w_\ell = 1$ such that $\mathbf{E}_{\mathbf{z} \sim \mathcal{N}_m}[g(\mathbf{z})p(\mathbf{z})] = 0$ holds for every odd polynomial $p$ of degree less than $k$, where $g(\mathbf{z}) = \sum_{\ell=1}^r w_\ell \text{sign}(\mathbf{v}_\ell^\intercal \mathbf{z})$. Noting that $\mathbf{E}_{\mathbf{z} \sim \mathcal{N}_m}[g(\mathbf{z})p(\mathbf{z})] = \sum_{\ell=1}^r w_\ell \mathbf{E}_{\mathbf{z} \sim \mathcal{N}_m}[p(\mathbf{z})\text{sign}(\mathbf{v}_\ell^\intercal \mathbf{z})]$, it suffices to bound the probability that there exist non-negative weights $w_1, \dots, w_r$ with $\sum_{\ell=1}^r w_\ell = 1$ such that $\sum_{\ell=1}^r w_\ell \mathbf{E}_{\mathbf{z} \sim \mathcal{N}_m}[p(\mathbf{z})\text{sign}(\mathbf{v}_\ell^\intercal \mathbf{z})] = 0$ holds for every odd polynomial $p$ of degree less than $k$. Our main technical lemma is the following:

**Lemma 3.5.** *Let* $p : \mathbb{R}^m \to \mathbb{R}$ *be a polynomial of degree less than* $k$ *and* $f \in \mathcal{L}^2(\mathbb{R}, \mathcal{N})$. *Let* $\mathbf{v} \in \mathbb{S}^{m-1}$ *be a unit vector. We have that* $\mathbf{E}_{\mathbf{z} \sim \mathcal{N}_m}[p(\mathbf{z})f(\mathbf{v}^\intercal \mathbf{z})]$ *is a polynomial in* $\mathbf{v}$ *of degree less than* $k$.

The proof proceeds by analyzing $p$ and $f$ as Hermite expansions. For completeness, we defer the proof of Lemma 3.5 to Appendix B. From Lemma 3.5, we can write $\sum_{\ell=1}^r w_\ell \mathbf{E}_{\mathbf{z} \sim \mathcal{N}_m}[p(\mathbf{z})\mathrm{sign}(\mathbf{v}_\ell^\mathsf{T} \mathbf{z})] = \sum_{\ell=1}^r w_\ell q(\mathbf{v}_\ell)$ for some odd polynomial $q$ of degree less than $k$ (since sign is odd). In this way, it suffices to show that with high probability there exist non-negative weights $w_1, \ldots, w_r$ with $\sum_{\ell=1}^r w_\ell = 1$ such that $\sum_{\ell=1}^r w_\ell q(\mathbf{v}_\ell) = 0$ for all odd polynomials $q$ of degree less than $k$. To prove this, we leverage the following lemma:

**Lemma 3.6.** *The following two statements are equivalent.*

1. *There exist non-negative weights $w_1, \ldots, w_r$ with $\sum_{\ell=1}^r w_\ell = 1$ such that $\sum_{\ell=1}^r w_\ell q(\mathbf{v}_\ell) = 0$ for all odd polynomials $q$ of degree less than $k$.*

2. *There does not exist any odd polynomial $q$ of degree less than $k$ such that $q(\mathbf{v}_\ell) > 0, 1 \leq \ell \leq r$.*

The proof of Lemma 3.6 follows via a careful application of LP duality; see Appendix B.

*Proof of Proposition 3.3.* Let $\Theta = \min_{1 \leq i < j \leq r} \arccos\langle \mathbf{v}_i, \mathbf{v}_j \rangle$. Applying Fact 3.4 by taking $\gamma = \left(\frac{1}{50\kappa_{m-1}}\right)^{\frac{1}{m-1}}$ yields $\mathbf{Pr}[\Theta \geq \Delta] \geq \mathbf{Pr}\left[\Theta \geq \gamma r^{-\frac{2}{m-1}}\right] \geq 99/100$ (since our choice of $m$ will imply $r = 1/\sqrt{50\kappa_{m-1}\Delta^{m-1}}$). This will imply that

$$\mathbf{Pr}\left[\min_{1 \leq i < j \leq r}\{\|\mathbf{v}_i - \mathbf{v}_j\|_2, \|\mathbf{v}_i + \mathbf{v}_j\|_2\} \geq \Omega(\Delta)\right] \geq 99/100 .$$

We then show that there exist non-negative weights $w_1, \ldots, w_r$ with $\sum_{\ell=1}^r w_\ell = 1$ such that $\mathbf{E}_{\mathbf{z} \sim \mathcal{N}_m}[g(\mathbf{z})p(\mathbf{z})] = 0$ holds for every odd polynomial $p$ with degree less than $k$, where $g(\mathbf{z}) = \sum_{\ell=1}^r w_\ell \mathrm{sign}(\mathbf{v}_\ell^\mathsf{T} \mathbf{z})$. By Lemma 3.5, it suffices to show that there exist non-negative weights $w_1, \ldots, w_r$ with $\sum_{\ell=1}^r w_\ell = 1$ such that $\sum_{\ell=1}^r w_\ell q(\mathbf{v}_\ell) = 0$ for all odd polynomials $q$ of degree less than $k$. By Fact 2.4, we have that $r \geq C\binom{m+k}{k} = C\mathrm{VC}(\mathcal{C}_k)$ for some sufficiently large constant $C > 0$. Note that for any odd polynomial $q$ in $m$ variables of degree less than $k$, we have that $\mathbf{E}[\mathrm{sign}(q(\mathbf{v}))] = 0$. Therefore, by the VC-inequality (Lemma 2.3), we have that

$$\mathbf{Pr}[\exists \text{ odd polynomial } q \text{ of degree less than } k \text{ such that } q(\mathbf{v}_\ell) > 0, 1 \leq \ell \leq r]$$
$$\leq \mathbf{Pr}\left[\sup_{f \in \mathcal{C}_{k-1}} \frac{\sum_{\ell=1}^r f(\mathbf{v}_\ell)}{r} = 1\right] \leq \mathbf{Pr}\left[\sup_{f \in \mathcal{C}_k} \left|\frac{\sum_{\ell=1}^r f(\mathbf{v}_\ell)}{r} - \mathbf{E}[f(\mathbf{v})]\right| \geq 1\right]$$
$$\leq \exp(-\Omega(r)) \leq 1/100.$$

Finally, applying Lemma 3.6 completes our proof. $\qquad\square$

## 3.2 Proof of SQ Lower Bound for General Weights

Let $k$ be a positive integer such that $r \geq C\binom{m+k}{k}$ for some constant $C > 0$ sufficiently large. Let $m = \frac{c \log r}{\log(1/\Delta)}$, where $r^{-1/10} \leq \Delta < 1$ and $1.99 \leq c \leq 2$. By Proposition 3.3, there exist unit vectors $\mathbf{v}_1, \ldots, \mathbf{v}_r \in \mathbb{R}^m$ such that

- There exist non-negative weights $w_1, \ldots, w_r$ with $\sum_{\ell=1}^r w_\ell = 1$ such that $\mathbf{E}_{\mathbf{z} \sim \mathcal{N}_m}[g(\mathbf{z})p(\mathbf{z})] = 0$ holds for every odd polynomial $p$ with degree less than $k$, where $g(\mathbf{z}) = \sum_{\ell=1}^r w_\ell \mathrm{sign}(\mathbf{v}_\ell^\mathsf{T} \mathbf{z})$.

- $\|\mathbf{v}_i + \mathbf{v}_j\|_2, \|\mathbf{v}_i - \mathbf{v}_j\|_2 \geq \Omega(\Delta), \forall 1 \leq i < j \leq r$ for some $\Delta > 0$.

For an arbitrary matrix $\mathbf{U} \in \mathbb{R}^{m \times n}$ with $\mathbf{U}\mathbf{U}^\mathsf{T} = \mathbf{I}_m$, denote by $D_\mathbf{U}$ the instance of mixture of linear classifiers with weight vectors $\mathbf{U}^\mathsf{T}\mathbf{v}_1, \ldots, \mathbf{U}^\mathsf{T}\mathbf{v}_r$ and weights $w_1, \ldots, w_r$. Let $\mathcal{D}_g = \{\mathcal{D}_\mathbf{U} \mid \mathbf{U} \in \mathbb{R}^{m \times n}, \mathbf{U}\mathbf{U}^\mathsf{T} = \mathbf{I}_m\}$. We have that $\|\mathbf{U}^\mathsf{T}\mathbf{v}_i \pm \mathbf{U}^\mathsf{T}\mathbf{v}_j\|_2 = \|\mathbf{v}_i \pm \mathbf{v}_j\|_2 \geq \Omega(\Delta), 1 \leq i < j \leq r$, and $\mathbf{E}_{(\mathbf{x},y) \sim D_\mathbf{U}}[y \mid \mathbf{x} = \mathbf{z}] = \sum_{\ell=1}^r w_\ell \mathrm{sign}((\mathbf{U}^\mathsf{T}\mathbf{v}_\ell)^\mathsf{T}\mathbf{z}) = \sum_{\ell=1}^r w_\ell \mathrm{sign}(\mathbf{v}_\ell^\mathsf{T}\mathbf{U}\mathbf{z}) = g(\mathbf{U}\mathbf{z}), \mathbf{z} \in \mathbb{R}^n$. By Proposition 3.2, any SQ algorithm for the decision problem $\mathcal{B}(\mathcal{D}_g, \mathcal{N}_n \times \mathcal{U}(\{\pm 1\}))$ must either use queries of tolerance $n^{-\Omega(k)}$, or make at least $2^{n^{\Omega(1)}}$ queries. The last step is to reduce the decision problem $\mathcal{B}(\mathcal{D}_g, \mathcal{N}_n \times \mathcal{U}(\{\pm 1\}))$ to the problem of learning mixture of linear classifiers.

**Claim 3.7** (see, e.g., Lemma 8.5 in [DK23]). *Suppose there exists an SQ algorithm to learn an unknown distribution in a family $\mathcal{D}$ to total variation distance $\epsilon$ using at most $N$ statistical queries*

*of tolerance $\tau$. Suppose furthermore that for each $D' \in \mathcal{D}$ we have that $d_{\mathrm{TV}}(D, D') > 2(\tau + \epsilon)$. Then there exists an SQ algorithm that solves the testing problem $\mathcal{B}(\mathcal{D}, D)$ using at most $n + 1$ queries of tolerance $\tau$.*

To apply Claim 3.7, we need to show that the distribution $D_\mathbf{U}$ in the class $\mathcal{D}_g$ is sufficiently far from the null hypothesis $D_0 = \mathcal{N}_n \times \mathcal{U}(\{\pm 1\})$ in total variation distance.

**Lemma 3.8.** *Let $\mathbf{U} \in \mathbb{R}^{m \times n}$ with $\mathbf{U}\mathbf{U}^\intercal = \mathbf{I}_m$. We have that $d_{\mathrm{TV}}(D_\mathbf{U}, D_0) \geq \Omega(\Delta/r)$.*

We briefly sketch the proof idea and defer the proof details to Appendix B. We consider $w_\ell \mathrm{sign}(\mathbf{v}_\ell^\intercal \mathbf{x})$ a halfspace of heaviest weight in $D_\mathbf{U}$, and pick points $\mathbf{z}$ and $\mathbf{z}'$ close to the defining hyperplane that are mirrors of each other over it. We note that under $D_0$ the expectations of $y$ conditioned on $\mathbf{x}$ being $\mathbf{z}$ or $\mathbf{z}'$ are both 0, whereas under $D_\mathbf{U}$ they likely (in particular, unless they are also on opposite sides of another halfspace) differ by at least $w_\ell$.

# 4 SQ Lower Bound for Uniform Mixtures via Spherical Designs

In this section, we prove our SQ lower bound for mixture of linear classifiers with uniform weights, thereby establishing Theorem 1.2. Our basic lower bound technique is essentially the same as in the previous section, but we need to construct a spherical design with *uniform weight*.

**Proposition 4.1.** *Let $m = \frac{c \log r}{\log(1/\Delta)}$, where $r^{-1/10} \leq \Delta < 1$ and $1.99 \leq c \leq 2$ is a constant. Let $k$ be an odd integer such that $r \geq \binom{m+2k-1}{2k}^5$. There exist vectors $\mathbf{v}_1, \ldots, \mathbf{v}_r$ on $\mathbb{S}^{m-1}$ such that:*

- $\mathbf{E}_{\mathbf{z} \sim \mathcal{N}_m}[g(\mathbf{z})p(\mathbf{z})] = 0$ *holds for every odd polynomial $p$ with degree less than $k$, where $g(\mathbf{z}) = \frac{1}{r}\sum_{\ell=1}^{r}\mathrm{sign}(\mathbf{v}_\ell^\intercal \mathbf{z})$.*

- $\|\mathbf{v}_i + \mathbf{v}_j\|_2, \|\mathbf{v}_i - \mathbf{v}_j\|_2 \geq \Omega(\Delta) - O\big(1/\binom{m+2k-1}{2k}\big), \forall 1 \leq i < j \leq r$.

By Lemma 3.5, we can write $\sum_{\ell=1}^{r} \mathbf{E}_{\mathbf{z} \sim \mathcal{N}_m}[p(\mathbf{z})\mathrm{sign}(\mathbf{v}_\ell^\intercal \mathbf{z})] = \sum_{\ell=1}^{r} q(\mathbf{v}_\ell)$ for some odd polynomial $q$ of degree less than $k$. Therefore, it suffices to show that with high probability there exist unit vectors $\mathbf{v}_1, \ldots, \mathbf{v}_r$ such that $\sum_{\ell=1}^{r} q(\mathbf{v}_\ell) = 0$ holds for every odd polynomial $q$ of degree less than $k$. To achieve this, we will leverage some techniques from [BRV13].

## 4.1 Spherical Design Construction

**Notation.** We start by introducing some additional notations we will use throughout this section. Let $\mathcal{P}_t^d$ denote the set of homogeneous polynomials in $d$ variables of some odd degree $t$. For any $p, q \in \mathcal{P}_t^d$, we consider the inner product $\langle p, q \rangle = \mathbf{E}_{\mathbf{x} \sim \mathcal{U}(\mathbb{S}^{d-1})}[p(\mathbf{x})q(\mathbf{x})]$. For any $p \in \mathcal{P}_t^d$, we define $\|p\|_2^2 = \langle p, p \rangle$. We denote by $N_{t,d} = \binom{t+d-1}{d-1}$ to be the dimension of $\mathcal{P}_t^d$ and $\Omega_t^d = \{p \in \mathcal{P}_t^d \mid \|p\|_2 \leq 1\}$. Let $\partial\Omega_t^d$ denote the boundary of $\Omega_t^d$, i.e., $\partial\Omega_t^d = \{p \in \mathcal{P}_t^d \mid \|p\|_2 = 1\}$. In the remaining part of this section, we will assume that the underlying distribution (over the expectation) is $\mathcal{U}(\mathbb{S}^{d-1})$.

The following theorem — a more detailed version of Theorem 1.6 — establishes the existence of a spherical $t$-design of size $\mathrm{poly}(N_{2t,d})$ with the desired pairwise separation properties.

**Theorem 4.2** (Spherical Design Construction). *Let $t$ be an odd integer and $r \geq N_{2t,d}^5$. Let $\mathbf{y}_1, \ldots, \mathbf{y}_r$ be uniform random vectors over $\mathbb{S}^{d-1}$. Then, with probability at least 99/100, there exist unit vectors $\mathbf{z}_1, \ldots, \mathbf{z}_r \in \mathbb{S}^{d-1}$ such that $\|\mathbf{z}_i - \mathbf{y}_i\|_2 \leq O(1/N_{2t,d}), i \in [r]$, and $(\mathbf{z}_1, \ldots, \mathbf{z}_r)$ form a spherical $t$-design.*

To prove Theorem 4.2, we will start from the following result.

**Theorem 4.3** ([BRV13]). *If there exists a continuous mapping $F : \mathcal{P}_t^d \to (\mathbb{S}^{d-1})^r$ such that for all $p \in \partial\Omega_t^d$, $\sum_{i=1}^{r} p(\mathbf{x}_i(p)) > 0$, where $F(p) = (\mathbf{x}_1(p), \ldots, \mathbf{x}_r(p))$, then there exists a polynomial $p^* \in \Omega_t^d$ such that $\mathbf{E}[p(\mathbf{x})] = \frac{1}{r}\sum_{i=1}^{r} p(\mathbf{x}_i(p^*))$ holds for every polynomial $p \in \mathcal{P}_t^d$.*

To apply Theorem 4.3, we need to find a continuous function $F$ mapping $\mathcal{P}_t^d$ to $(\mathbb{S}^{d-1})^r$ such that for any $p \in \partial\Omega_t^d$, $\sum_{i=1}^{r} p(\mathbf{z}_i) > 0$, where $F(p) = (\mathbf{z}_1, \ldots, \mathbf{z}_r)$. We will construct the mapping

$F$ as follows: we sample $\mathbf{y}_1, \ldots, \mathbf{y}_r$ uniformly over the unit sphere $\mathbb{S}^{d-1}$, and then try to make the value of $p(\mathbf{y}_i)$ larger by moving each point $\mathbf{y}_i$ in the direction of the gradient. In particular, we let $\mathbf{z}_i = \frac{\mathbf{y}_i + \delta \nabla_o p(\mathbf{y}_i)}{\|\mathbf{y}_i + \delta \nabla_o p(\mathbf{y}_i)\|_2}$ for some $\delta > 0$ sufficiently small, where $\nabla_o p(\mathbf{y})$ is the component of $\nabla p(\mathbf{y})$ orthogonal to the direction $\mathbf{y}$. We will prove that with high probability, for any $p \in \partial\Omega_t^d$, $\sum_{i=1}^r p(\mathbf{z}_i) > 0$. Intuitively, this works because of two facts:

1. With high probability over the choice of $\mathbf{y}_i$, for all $p \in \Omega_t^d$, the average value of $p(\mathbf{y})$ is already close to zero.

2. Moving in the direction of $\nabla_o p(\mathbf{y}_i)$ increases $p(\mathbf{y}_i)$ by a notable amount.

**Lemma 4.4.** *Let $\Omega$ be a subspace of polynomials in $d$ variables with mean zero. Let $N$ be the dimension of $\Omega$. Let $\mathbf{x}_1, \ldots, \mathbf{x}_r$ be i.i.d. random vectors over $\mathbb{S}^{d-1}$. Then, with probability at least $1 - \frac{N}{r\eta^2}$, we have that for any $p \in \Omega$, $\left| \frac{1}{r} \sum_{i=1}^r p(\mathbf{x}_i) \right| \leq \eta \|p\|_2$.*

To prove Lemma 4.4, we take an orthonormal basis $p_1, \ldots, p_N \in \Omega$ and define $\mathbf{p}(\mathbf{x}) \stackrel{\text{def}}{=} [p_1(\mathbf{x}), \ldots, p_N(\mathbf{x})]^\intercal$. Noting that $\mathbf{p}(\mathbf{x})$ is a random vector with mean zero and covariance identity, applying Markov's inequality will yield the result. We defer the proof details to Appendix C.

**Lemma 4.5.** *Let $\mathbf{y} \in \mathbb{S}^{d-1}$ and $0 < \delta \leq 1/N_{2t,d}^2$. For any $p \in \partial\Omega_t^d$, let $\mathbf{z} = \frac{\mathbf{y} + \delta \nabla_o p(\mathbf{y})}{\|\mathbf{y} + \delta \nabla_o p(\mathbf{y})\|_2}$. We have that $p(\mathbf{z}) - p(\mathbf{y}) \geq C\delta \|\nabla_o p(\mathbf{y})\|_2^2$ for some universal constant $0 < C < 1$.*

The proof of Lemma 4.5 follows by Taylor expansion, where the contributions to $p(\mathbf{z})$ coming from higher-order terms will not be large as long as $\mathbf{z}$ is sufficiently close to $\mathbf{y}$. We defer the proof details to Appendix C. By applying the above two lemmas, we establish the following:

**Theorem 4.6.** *Let $\mathbf{y}_1, \ldots, \mathbf{y}_r$ be i.i.d. random vectors over $\mathbb{S}^{d-1}$. Let $\delta = 1/N_{2t,d}^2$ and $r \geq N_{2t,d}^5$. We consider the mapping $F : \mathcal{P}_t^d \to (\mathbb{S}^{d-1})^r$ as follows: for any $p \in \mathcal{P}_t^d$, let*

$$\mathbf{z}_i = \frac{\mathbf{y}_i + \delta \nabla_o p(\mathbf{y}_i)}{\|\mathbf{y}_i + \delta \nabla_o p(\mathbf{y}_i)\|_2} \ ,$$

*where $\nabla_o p(\mathbf{y})$ is the component of $\nabla p(\mathbf{y})$ orthogonal to the direction $\mathbf{y}$. Let $F(p) := (\mathbf{z}_1, \ldots, \mathbf{z}_r)$. Then, with probability at least $99/100$, we have that for any $p \in \partial\Omega_t^d$, $\sum_{i=1}^r p(\mathbf{z}_i) > 0$.*

To prove Theorem 4.6, we consider $\widetilde{p}(\mathbf{y}) = p(\mathbf{y}) + C\delta(\|\nabla_o p(\mathbf{y})\|_2^2 - \mathbf{E}[\|\nabla_o p(\mathbf{y})\|_2^2])$, where $C, \delta$ comes from Lemma 4.5. Noting that $\mathbf{E}[\widetilde{p}(\mathbf{y})] = 0$ and $\widetilde{p}(\mathbf{y})$ is a polynomial containing only monomials of degree $2t, 2t-2, t, 0$, we are able to apply Lemma 4.4 to obtain the desired result. See Appendix C for the proof.

We now prove our main technical result, Theorem 4.2.

*Proof of Theorem 4.2.* Let $\delta = 1/N_{2t,d}^2$. We consider the mapping $F : \mathcal{P}_t^d \to (\mathbb{S}^{d-1})^r$ as follows: for any $p \in \mathcal{P}_t^d$, let $\mathbf{z}_i = \frac{\mathbf{y}_i + \delta \nabla_o p(\mathbf{y}_i)}{\|\mathbf{y}_i + \delta \nabla_o p(\mathbf{y}_i)\|_2}$, where $\nabla_o p(\mathbf{y})$ is the component of $\nabla p(\mathbf{y})$ orthogonal to the direction $\mathbf{y}$. By Theorem 4.6, with probability at least $99/100$, we have that for any $p \in \partial\Omega_t^d$, $\sum_{i=1}^r p(\mathbf{z}_i) > 0$. Applying Theorem 4.3 yields that there exists some $p^* \in \Omega_t^d$ such that $F(p^*) = (\mathbf{z}_1^*, \ldots, \mathbf{z}_r^*)$ form a spherical $t$-design. Furthermore, we can show that $\|\mathbf{z}_i^* - \mathbf{y}_i\|_2 \leq O(1/N_{2t,d})$ by elementary calculation (see Appendix C). $\square$

## 4.2 Proof of Theorem 1.2

We first prove Proposition 4.1 based on our construction of spherical $t$-design. To achieve this, it suffices to show that with high probability there exist $\mathbf{v}_1, \ldots, \mathbf{v}_r \in \mathbb{S}^{m-1}$ such that

- $\mathbf{v}_1, \ldots, \mathbf{v}_r$ is a spherical $k$-design.
- $\|\mathbf{v}_i + \mathbf{v}_j\|_2, \|\mathbf{v}_i - \mathbf{v}_j\|_2 \geq \Omega(\Delta), \forall 1 \leq i < j \leq r$ for some $\Delta > 0$.

*Proof of Proposition 4.1.* Let $\Theta = \min_{1 \leq i < j \leq r} \arccos\langle \mathbf{y}_i, \mathbf{y}_j \rangle$. Applying Fact 3.4 by taking $\gamma = \left(\frac{1}{50\kappa_{m-1}}\right)^{\frac{1}{m-1}}$ yields $\mathbf{Pr}[\Theta \geq \Delta] \geq \mathbf{Pr}\left[\Theta \geq \gamma r^{-\frac{2}{m-1}}\right] \geq 99/100$. This will give

$$\mathbf{Pr}\left[\min_{1 \leq i < j \leq r}\{\|\mathbf{y}_i - \mathbf{y}_j\|_2, \|\mathbf{y}_i + \mathbf{y}_j\|_2\} \geq \Omega(\Delta)\right] \geq 99/100 \ ,$$

since our choice of $m$ will imply that $r = 1/\sqrt{50\kappa_{m-1}\Delta^{m-1}}$. By Theorem 4.2, with probability at least 99/100, there exist unit vectors $\mathbf{z}_1^*, \ldots, \mathbf{z}_r^* \in \mathbb{S}^{m-1}$ such that $\|\mathbf{z}_i^* - \mathbf{y}_i\|_2 \le O(1/N_{2k,m}), i \in [r]$ and $(\mathbf{z}_1^*, \ldots, \mathbf{z}_r^*)$ form a spherical $k$-design. Therefore, for any odd homogeneous polynomial $p$ in $m$ variables of odd degree $t < k$, we have that

$$\frac{1}{r}\sum_{i=1}^{r} p(\mathbf{z}_i^*) = \frac{1}{r}\sum_{i=1}^{r}(\|\mathbf{z}_i^*\|_2^2)^{\frac{k-t}{2}} p(\mathbf{z}_i^*) = \mathbf{E}\left[(\|\mathbf{z}\|_2^2)^{\frac{k-t}{2}} p(\mathbf{z})\right] = 0\ ,$$

which implies that $\sum_{i=1}^{r} q(\mathbf{z}_i^*) = 0$ holds for any polynomial $q$ in $m$ variables of degree less than $k$. In addition, for every $1 \le i < j \le r$, we have that

$$\begin{aligned}
\|\mathbf{z}_i^* \pm \mathbf{z}_j^*\|_2 &= \|(\mathbf{y}_i \pm \mathbf{y}_j) + (\mathbf{z}_i^* - \mathbf{y}_i) \pm (\mathbf{z}_j^* - \mathbf{y}_j)\|_2 \ge \|\mathbf{y}_i \pm \mathbf{y}_j\| - \|\mathbf{z}_i^* - \mathbf{y}_i\|_2 - \|\mathbf{z}_j^* - \mathbf{y}_j\|_2 \\
&\ge \|\mathbf{y}_i \pm \mathbf{y}_j\|_2 - O(1/N_{2k,m})\ .
\end{aligned}$$

This completes the proof. $\qquad\square$

*Proof of Theorem 1.2.* Let $m = \frac{c\log r}{\log(1/\Delta)}$, where $r^{-1/10} \le \Delta < 1$ and $1.99 \le c \le 2$ is a constant. Let $c' = \frac{(1/e)(1/\Delta)^{1/(5c)}-1}{2}$ and $k = c'm = \frac{\left((1/e)(1/\Delta)^{1/(5c)}-1\right)c\log r}{2\log(1/\Delta)}$. In this way, we will have that $r \ge N_{2k,m}^5$ (see Appendix C for calculation details). Therefore, by Proposition 4.1, there exist vectors $\mathbf{v}_1, \ldots, \mathbf{v}_r \in \mathbb{S}^{m-1}$ such that

- $\mathbf{E}_{\mathbf{z}\sim\mathcal{N}_m}[g(\mathbf{z})p(\mathbf{z})] = 0$ holds for every odd polynomial $p$ of degree less than $k$, where $g(\mathbf{z}) = \frac{1}{r}\sum_{\ell=1}^{r}\mathrm{sign}(\mathbf{v}_\ell^\intercal\mathbf{z})$.

- $\|\mathbf{v}_i + \mathbf{v}_j\|_2, \|\mathbf{v}_i - \mathbf{v}_j\|_2 \ge \Omega(\Delta) - O(1/N_{2k,m}), \forall 1 \le i < j \le r$.

For an arbitrary matrix $\mathbf{U} \in \mathbb{R}^{m\times n}$ with $\mathbf{U}\mathbf{U}^\intercal = \mathbf{I}_m$, denote by $D_\mathbf{U}$ the instance of mixture of linear classifiers with weight vectors $\mathbf{U}^\intercal\mathbf{v}_1, \ldots, \mathbf{U}^\intercal\mathbf{v}_r$. Let $\mathcal{D}_g = \{\mathcal{D}_\mathbf{U} \mid \mathbf{U} \in \mathbb{R}^{m\times n}, \mathbf{U}\mathbf{U}^\intercal = \mathbf{I}_m\}$. By definition, we have that $\mathbf{E}_{(\mathbf{x},y)\sim D_\mathbf{U}}[y \mid \mathbf{x} = \mathbf{z}] = \frac{1}{r}\sum_{\ell=1}^{r}\mathrm{sign}((\mathbf{U}^\intercal\mathbf{v}_\ell)^\intercal\mathbf{z}) = \frac{1}{r}\sum_{\ell=1}^{r}\mathrm{sign}(\mathbf{v}_\ell^\intercal\mathbf{U}\mathbf{z}) = g(\mathbf{U}\mathbf{z}), \mathbf{z} \in \mathbb{R}^n$, and $\|\mathbf{U}^\intercal\mathbf{v}_i\pm\mathbf{U}^\intercal\mathbf{v}_j\|_2 = \|\mathbf{v}_i\pm\mathbf{v}_j\|_2 \ge \Omega(\Delta), 1 \le i < j \le r$, since $N_{2k,m} \ge \Omega((1/\Delta)^{1.89})$ (see Appendix C for the calculation details). Therefore by Proposition 3.2, any SQ algorithm that solves the decision problem $\mathcal{B}(\mathcal{D}_g, \mathcal{N}_n \times \mathcal{U}(\{\pm 1\}))$ must either use queries of tolerance $n^{-\Omega(k)}$, or make at least $2^{n^{\Omega(1)}}$ queries. By Lemma 3.8, for any $\mathbf{U} \in \mathbb{R}^{m\times n}$ with $\mathbf{U}\mathbf{U}^\intercal = \mathbf{I}_m$, we have that $d_{\mathrm{TV}}(D_\mathbf{U}, D_0) \ge \Omega(\Delta/r) \ge 2(n^{-\Omega(k)}+\epsilon)$, where $D_0 = \mathcal{N}_n \times \mathcal{U}(\{\pm 1\})$. Therefore, by Claim 3.7, any SQ algorithm that learns a distribution in $\mathcal{D}_g$ within error $\epsilon$ in total variation distance must either use queries of tolerance $n^{-\Omega(\log r/(\Delta^{1/(5c)}\log(1/\Delta)))}$, or make at least $2^{n^{\Omega(1)}}$ queries. This completes the proof. $\qquad\square$

## 5  Conclusion

This work establishes a near-optimal Statistical Query (SQ) lower bound for learning uniform mixtures of linear classifiers under the Gaussian distribution. Our lower bound nearly matches prior algorithmic work on the problem [CDV22]. Our result applies for the simplest (and well-studied) distributional setting where the covariates are drawn from the standard Gaussian distribution. This directly implies similar information-computation tradeoffs for the setting that the covariates are drawn from a more general distribution family (e.g., an unknown subgaussian or log-concave distribution) that includes the standard normal.

From a technical perspective, we believe that our new efficient construction of spherical designs is a mathematical contribution of independent interest that could be used to establish SQ lower bounds for other related latent variable models (e.g., various mixtures of experts). A natural direction is to establish information-computation tradeoffs for a fixed non-Gaussian distribution on covariates (e.g., the uniform distribution over the Boolean hypercube), for which a different hardness construction is needed.

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
