## Supplementary Material

## A    Background on Hermite Polynomials

Recall the definition of the probabilist's Hermite polynomials:

$$He_n(x) = (-1)^n e^{x^2/2} \cdot \frac{d^2}{dx^2} e^{-x^2/2}.$$

Under this definition, the first four Hermite polynomials are

$$He_0(x) = 1, He_1(x) = x, He_2(x) = x^2 - 1, He_3(x) = x^3 - 3x.$$

In our work, we will consider the *normalized* Hermite polynomial of degree $n$ to be $h_n(x) = \frac{He_n(x)}{\sqrt{n!}}$. These normalized Hermite polynomials form a complete orthogonal basis for inner product space $\mathcal{L}^2(\mathbb{R}, \mathcal{N})$. To obtain an orthogonal basis for $\mathcal{L}^2(\mathbb{R}^d, \mathcal{N}_d)$, we will use a multi-index $J = (j_1, \ldots, j_d) \in \mathbb{N}^d$ to define the $d$-variate normalized Hermite polynomial as $H_J(\mathbf{x}) = \prod_{i=1}^d H_{j_i}(x_i)$. Let the total degree of $H_J$ be $|J| = \sum_{i=1}^d j_i$. Given a function $f \in \mathcal{L}^2(\mathbb{R}^d, \mathcal{N}_d)$, we can express it uniquely as $f(\mathbf{x}) = \sum_{J \in \mathbb{N}^d} \widehat{f}(J) H_J(\mathbf{x})$, where $\widehat{f}(J) = \mathbf{E}_{\mathbf{x} \in \mathcal{N}_d}[f(\mathbf{x}) H_J(\mathbf{x})]$. We denote by $f^{[k]}(\mathbf{x})$ the degree $k$ part of the Hermite expansion of $f$, i.e., $f^{[k]}(\mathbf{x}) = \sum_{|J|=k} \widehat{f}(J) H_J(\mathbf{x})$.

**Definition A.1.** We say that a polynomial $q$ in $d$ variables is harmonic of degree $k$ if it is a linear combination of degree $k$ Hermite polynomials. That is, $q$ is harmonic if it can be written as

$$q(\mathbf{x}) = q^{[k]}(\mathbf{x}) = \sum_{J:|J|=k} c_J H_J(\mathbf{x}).$$

Notice that, since for a single-dimensional Hermite polynomial it holds $h'_m(x) = \sqrt{m} h_{m-1}(x)$, we have that $\nabla H_M^{(i)}(\mathbf{x}) = \sqrt{m_i} H_{M-E_i}(\mathbf{x})$, where $M = (m_1, \ldots, m_d)$. From this fact and the orthogonality of Hermite polynomials, we obtain

$$\mathbf{E}_{\mathbf{x} \sim \mathcal{N}_d}[\langle \nabla H_M(\mathbf{x}), \nabla H_L(\mathbf{x}) \rangle] = |M| \, \mathbb{I}[M = L] \,.$$

We will also require the following standard facts:

**Fact A.2.** *Let $p$ be a polynomial of degree $k$ in $d$ variables. Then $p$ is harmonic of degree $k$ if and only if for all $\mathbf{x} \in \mathbb{R}^d$ it holds that $kp(\mathbf{x}) = \langle \mathbf{x}, \nabla p(\mathbf{x}) \rangle - \nabla^2 p(\mathbf{x})$.*

**Fact A.3** (see, e.g., [DKPZ21])**.** *Let $p, q$ be harmonic polynomials of degree $k$. Then,*

$$\mathbf{E}_{\mathbf{x} \sim \mathcal{N}_d}\left[\langle \nabla^\ell p(\mathbf{x}), \nabla^\ell q(\mathbf{x}) \rangle\right] = k(k-1) \ldots (k - \ell + 1) \mathbf{E}_{\mathbf{x} \sim \mathcal{N}_d}[p(\mathbf{x}) q(\mathbf{x})].$$

*In particular,*

$$\langle \nabla^k p(\mathbf{x}), \nabla^k q(\mathbf{x}) \rangle = k! \mathbf{E}_{\mathbf{x} \sim \mathcal{N}_d}[p(\mathbf{x}) q(\mathbf{x})].$$

## B    Omitted Proofs from Section 3

### B.1    Proof of Lemma 3.5

We start with the following claim:

**Claim B.1.** *Let $p : \mathbb{R}^{n_1} \to \mathbb{R}$ and $q : \mathbb{R}^{n_2} \to \mathbb{R}$, where $p$ is a polynomial of degree at most $k$ and $q \in \mathcal{L}^2(\mathbb{R}^{n_2}, \mathcal{N}_{n_2})$. Let $\mathbf{U} \in \mathbb{R}^{n_1 \times n}, \mathbf{V} \in \mathbb{R}^{n_2 \times n}$ such that $\mathbf{U}\mathbf{U}^\intercal = \mathbf{I}_{n_1}, \mathbf{V}\mathbf{V}^\intercal = \mathbf{I}_{n_2}$. Then, we have that $\mathbf{E}_{\mathbf{x} \sim \mathcal{N}_n}[p(\mathbf{U}\mathbf{x})q(\mathbf{V}\mathbf{x})] = \sum_{m=0}^k \frac{1}{m!} \langle (\mathbf{U}^\intercal)^{\otimes m} \mathbf{R}_1^m, (\mathbf{V}^\intercal)^{\otimes m} \mathbf{R}_2^m \rangle$, where $\mathbf{R}_1^m = \nabla^m p^{[m]}(\mathbf{x}), \mathbf{R}_2^m = \nabla^m q^{[m]}(\mathbf{x})$.*

We require the following lemma:

**Lemma B.2.** *Let $p$ be a harmonic polynomial of degree $k$. Let $\mathbf{V} \in \mathbb{R}^{m \times n}$ with $\mathbf{V}\mathbf{V}^\intercal = \mathbf{I}_m$. Then the polynomial $p(\mathbf{V}\mathbf{x})$ is harmonic of degree $k$.*

*Proof.* Let $f(\mathbf{x}) = p(\mathbf{Vx})$. By Fact A.2, it suffices to show that for all $\mathbf{x} \in \mathbb{R}^n$ it holds that $kf(\mathbf{x}) = \langle \mathbf{x}, \nabla f(\mathbf{x}) \rangle - \nabla^2 f(\mathbf{x})$. Since $\mathbf{VV}^\mathsf{T} = \mathbf{I}_m$, applying Fact A.2 yields

$$\langle \mathbf{x}, \nabla f(\mathbf{x}) \rangle - \nabla^2 f(\mathbf{x}) = \langle \mathbf{Vx}, \nabla p(\mathbf{Vx}) \rangle - \nabla^2 p(\mathbf{Vx}) = kp(\mathbf{Vx}) = kf(\mathbf{x}) \,.$$

$\square$

*Proof of Claim B.1.* For $m \in \mathbb{N}$, let $f^{(m)}(\mathbf{x}) = p^{[m]}(\mathbf{Ux})$ and $g^{(m)}(\mathbf{x}) = q^{[m]}(\mathbf{Vx})$. We can write $p(\mathbf{Ux}) \sim \sum_{m=0}^{k} f^{(m)}(\mathbf{x})$ and $q(\mathbf{Vx}) \sim \sum_{m=0}^{\infty} g^{(m)}(\mathbf{x})$. Then applying Fact A.3 and Lemma B.2 yields

$$\mathbf{E}_{\mathbf{x} \sim \mathcal{N}_n}[p(\mathbf{Ux})q(\mathbf{Vx})] = \sum_{m_1=0}^{k} \sum_{m_2=0}^{\infty} \mathbf{E}_{\mathbf{x} \sim \mathcal{N}_n}[f^{(m_1)}(\mathbf{x})g^{(m_2)}(\mathbf{x})] = \sum_{m=0}^{k} \mathbf{E}_{\mathbf{x} \sim \mathcal{N}_n}[f^{(m)}(\mathbf{x})g^{(m)}(\mathbf{x})]$$

$$= \sum_{m=0}^{k} \frac{1}{m!} \left\langle \nabla^m f^{(m)}(\mathbf{x}), \nabla^m g^{(m)}(\mathbf{x}) \right\rangle = \sum_{m=0}^{k} \frac{1}{m!} \left\langle \nabla^m p^{[m]}(\mathbf{Ux}), \nabla^m q^{[m]}(\mathbf{Vx}) \right\rangle \,.$$

Denote by $\mathcal{U} \subseteq \mathbb{R}^n$ the image of the linear map $\mathbf{U}^\mathsf{T}$. Applying the chain rule, for any function $h(\mathbf{Ux}) : \mathbb{R}^n \to \mathbb{R}$, it holds $\nabla h(\mathbf{Ux}) = \partial_i h(\mathbf{Ux})U_{ij} \in \mathcal{U}$, where we applied Einstein's summation notation for repeated indices. Applying the above rule $m$ times, we have that

$$\nabla^m h(\mathbf{Ux}) = \partial_{i_m} \ldots \partial_{i_1} h(\mathbf{Ux})U_{i_1,j_1} \ldots U_{i_m,j_m} \in \mathcal{U}^{\otimes m}.$$

Moreover, denote $\mathbf{S}_m = \nabla^m p^{[m]}(\mathbf{Ux}) = (\mathbf{U}^\mathsf{T})^{\otimes m}\mathbf{R}_1^m \in \mathcal{U}^{\otimes m}$, and $\mathbf{T}_m = \nabla^m q^{[m]}(\mathbf{Vx}) = (\mathbf{V}^\mathsf{T})^{\otimes m}\mathbf{R}_2^m \in \mathcal{V}^{\otimes m}$. We have that

$$\mathbf{E}_{\mathbf{x} \sim \mathcal{N}_n}[f(\mathbf{x})g(\mathbf{x})] = \sum_{m=0}^{k} \frac{1}{m!} \left\langle \nabla^m p^{[m]}(\mathbf{Ux}), \nabla^m q^{[m]}(\mathbf{Vx}) \right\rangle = \sum_{m=0}^{k} \frac{1}{m!} \langle \mathbf{S}_m, \mathbf{T}_m \rangle$$

$$= \sum_{m=0}^{k} \frac{1}{m!} \langle (\mathbf{U}^\mathsf{T})^{\otimes m}\mathbf{R}_1^m, (\mathbf{V}^\mathsf{T})^{\otimes m}\mathbf{R}_2^m \rangle.$$

This proves the claim. $\square$

*Proof of Lemma 3.5.* Applying Claim B.1 by taking $\mathbf{U} = \mathbf{I}_m$ and $\mathbf{V} = \mathbf{v}^\mathsf{T}$, we have that

$$\mathbf{E}_{\mathbf{z} \sim \mathcal{N}_m}\left[p(\mathbf{z})f(\mathbf{v}^\mathsf{T}\mathbf{z})\right] = \sum_{d=0}^{k-1} \frac{1}{d!} \langle \mathbf{R}_1^d, \mathbf{v}^{\otimes d}\mathbf{R}_2^d \rangle,$$

which is a polynomial in $\mathbf{v}$ of degree less than $k$, since $\mathbf{R}_1^d = \nabla^d p^{[d]}(\mathbf{x})$ and $\mathbf{R}_2^d = \nabla^d f^{[d]}(\mathbf{x})$ are constants only depending on $p$ and $f$. This completes the proof of Lemma 3.5. $\square$

### B.2 Proof of Lemma 3.6

We start by proving that *"there exist non-negative weights $w_1, \ldots, w_r$ with $\sum_{\ell=1}^{r} w_\ell = 1$ such that $\sum_{\ell=1}^{r} w_\ell q(\mathbf{v}_\ell) = 0$ for all odd polynomials $q$ of degree less than $k$"* implies *"there does not exist any odd polynomial $q$ of degree less than $k$ such that $q(\mathbf{v}_\ell) > 0, 1 \leq \ell \leq r$."* Suppose for contradiction that there exists an odd polynomial $q^*$ of degree less than $k$ such that $q^*(\mathbf{v}_\ell) > 0, 1 \leq \ell \leq r$. For arbitrary non-negative weights $w_1, \ldots, w_r$ with $\sum_{\ell=1}^{r} w_\ell = 1$, we have that $\sum_{\ell=1}^{r} w_\ell q^*(\mathbf{v}_\ell) \geq \min\{q^*(\mathbf{v}_1), \ldots, q^*(\mathbf{v}_r)\} > 0$, which contradicts to the first statement.

We then prove the opposite direction. We will use the following version of Farkas' lemma.

**Fact B.3** (Farkas' lemma). *Let $A \in \mathbb{R}^{m \times n}$ and $b \in \mathbb{R}^m$. Then exactly one of the following two assertions is true:*

- *There exists an $\mathbf{x} \in \mathbb{R}^n$ such that $A\mathbf{x} = b$ and $\mathbf{x} \geq 0$.*

- *There exists a $\mathbf{y} \in \mathbb{R}^m$ such that $\mathbf{y}^\mathsf{T}A \geq 0$ and $\mathbf{y}^\mathsf{T}b < 0$.*

Suppose for contradiction that there does not exist $w_1, \ldots, w_r$ with $\sum_{\ell=1}^{r} w_\ell = 1$ such that $\sum_{\ell=1}^{r} w_\ell q(\mathbf{v}_\ell) = 0$ holds for every odd polynomial $q$ of degree less than $k$. Let $s_{k,m}$ denote the total number of $m$-variate odd monomials of degree less than $k$, and $\{q_j^{k,m}\}_{1 \leq j \leq s_{k,m}}$ denote such monomials. We consider the following LP with variables $\mathbf{w} = (w_1, \ldots, w_r)^{\mathsf{T}}$: $\sum_{\ell=1}^{r} w_\ell q_j^{k,m}(\mathbf{v}_\ell) = 0, 1 \leq j \leq s_{k,m}, \sum_{\ell=1}^{r} w_\ell = 1, w_\ell \geq 0, 1 \leq \ell \leq r$. By our assumption, the LP is infeasible. In order to applying the Farkas Lemma (Fact B.3), we write the linear system as $A\mathbf{w} = b$, where

$$
\mathbf{A} = \begin{bmatrix} 1 & 1 & \cdots & 1 \\ q_1^{k,m}(\mathbf{v}_1) & q_1^{k,m}(\mathbf{v}_2) & \cdots & q_1^{k,m}(\mathbf{v}_r) \\ \vdots & \vdots & \ddots & \vdots \\ q_{s_{k,m}}^{k,m}(\mathbf{v}_1) & q_{s_{k,m}}^{k,m}(\mathbf{v}_2) & \cdots & q_{s_{k,m}}^{k,m}(\mathbf{v}_r) \end{bmatrix}, \mathbf{w} = \begin{bmatrix} w_1 \\ w_2 \\ \vdots \\ w_r \end{bmatrix}, \mathbf{b} = \begin{bmatrix} 1 \\ 0 \\ \vdots \\ 0 \end{bmatrix}.
$$

By Fact B.3, the original linear system is infeasible if and only if there exists a vector $\mathbf{u} = [u_0, u_1, \ldots, u_{s_{k,d}}]^{\mathsf{T}}$, $\mathbf{u}^{\mathsf{T}}\mathbf{A} \geq 0$ and $\mathbf{u}^{\mathsf{T}}\mathbf{b} < 0$, which is equivalent to $u_0 + \sum_{j=1}^{s_{k,m}} u_j q_j^{k,m}(\mathbf{v}_\ell) \geq 0, \forall 1 \leq \ell \leq r$ and $u_0 < 0$. Let $q^*(\mathbf{v}) = \sum_{j=1}^{s_{k,m}} u_j q_j^{k,m}(\mathbf{v}), \mathbf{v} \in \mathbb{R}^m$, which is an odd polynomial of degree less than $k$. By our definition of $q^*$, we have that $q^*(\mathbf{v}_\ell) = \sum_{j=1}^{s_{k,m}} u_j q_j^{k,m}(\mathbf{v}_\ell) \geq -u_0 > 0, \forall 1 \leq \ell \leq r$, which contradicts to our assumption that *there does not exist any odd polynomial $q$ of degree less than $k$ such that $q(\mathbf{v}_\ell) > 0, \forall 1 \leq \ell \leq r$*. This completes the proof.

### B.3 Proof of Claim 3.7

We denote by $G(\mathbf{x})$ to be the standard Gaussian density. By definition, we have that

$$
d_{\mathrm{TV}}(D_{\mathbf{U}}, D_0) = (1/2) \int_{\mathbf{x} \in \mathbb{R}^n} \sum_{y \in \{\pm 1\}} |D_{\mathbf{U}}(\mathbf{x}, y) - D_0(\mathbf{x}, y)| d\mathbf{x}
$$

$$
= (1/2) \int_{\mathbf{x} \in \mathbb{R}^n} G(\mathbf{x}) \sum_{y \in \{\pm 1\}} \left| \sum_{\ell=1}^{r} w_\ell \mathbb{I}[\mathrm{sign}(\mathbf{v}_\ell^{\mathsf{T}} \mathbf{U} \mathbf{x}) = y] - (1/2) \right| d\mathbf{x}
$$

$$
= (1/2) \mathbf{E}_{\mathbf{x} \sim \mathcal{N}_n} \left[ \sum_{y \in \{\pm 1\}} \left| \sum_{\ell=1}^{r} w_\ell \mathbb{I}[\mathrm{sign}(\mathbf{v}_\ell^{\mathsf{T}} \mathbf{U} \mathbf{x}) = y] - (1/2) \right| \right]
$$

$$
= (1/2) \sum_{y \in \{\pm 1\}} \mathbf{E}_{\mathbf{x} \sim \mathcal{N}_n} \left[ \left| \sum_{\ell=1}^{r} w_\ell \mathbb{I}[\mathrm{sign}(\mathbf{v}_\ell^{\mathsf{T}} \mathbf{U} \mathbf{x}) = y] - (1/2) \right| \right].
$$

Therefore, it suffices to show that

$$
\mathbf{E}_{\mathbf{x} \sim \mathcal{N}_n} \left[ \left| \sum_{\ell=1}^{r} w_\ell \mathbb{I}[\mathrm{sign}(\mathbf{v}_\ell^{\mathsf{T}} \mathbf{U} \mathbf{x}) = y] - (1/2) \right| \right] \geq \Omega(\Delta/r), \quad \forall y \in \{\pm 1\}.
$$

We assume that $w_{\ell_0} \geq 1/r$ for some $\ell_0 \in [r]$. Let $\mathbf{v}^*$ be an arbitrary vector satisfying $\mathbf{v}_{\ell_0}^{\mathsf{T}} \mathbf{v}^* = 0$. We denote by

$$
\mathcal{X}_1 = \{\mathbf{x} \in \mathbb{R}^m \mid \mathrm{sign}(\mathbf{v}_{\ell_0}^{\mathsf{T}} \mathbf{x}) > 0, \mathrm{sign}(\mathbf{v}_\ell^{\mathsf{T}} \mathbf{x}) = \mathrm{sign}(\mathbf{v}_\ell^{\mathsf{T}} \mathbf{v}^*), \ell \in [r] \setminus \{\ell_0\}\},
$$

$$
\mathcal{X}_2 = \{\mathbf{x} \in \mathbb{R}^m \mid \mathrm{sign}(\mathbf{v}_{\ell_0}^{\mathsf{T}} \mathbf{x}) < 0, \mathrm{sign}(\mathbf{v}_\ell^{\mathsf{T}} \mathbf{x}) = \mathrm{sign}(\mathbf{v}_\ell^{\mathsf{T}} \mathbf{v}^*), \ell \in [r] \setminus \{\ell_0\}\}.
$$

Roughly speaking, $\mathcal{X}_1$ and $\mathcal{X}_2$ denote the subsets of vectors which are very close to the boundary of the halfspace with direction $\mathbf{v}_{\ell_0}$ and maintain the same label with the boundary for the other halfspaces. By definition, for any $\mathbf{x}_1 \in \mathcal{X}_1, \mathbf{x}_2 \in \mathcal{X}_2$, we have that

$$
\left| \sum_{\ell=1}^{r} w_\ell \mathbb{I}[\mathrm{sign}(\mathbf{v}_\ell^{\mathsf{T}} \mathbf{x}_1) = y] - \sum_{\ell=1}^{r} w_\ell \mathbb{I}[\mathrm{sign}(\mathbf{v}_\ell^{\mathsf{T}} \mathbf{x}_2) = y] \right| = w_{\ell_0} \geq 1/r.
$$

Therefore, we have either

$$
\left| \sum_{\ell=1}^{r} w_\ell \mathbb{I}[\mathrm{sign}(\mathbf{v}_\ell^{\mathsf{T}} \mathbf{x}_1) = y] - (1/2) \right| \geq 1/2r, \quad \forall \mathbf{x}_1 \in \mathcal{X}_1,
$$

or

$$\left|\sum_{\ell=1}^{r} w_\ell \mathbb{I}[\text{sign}(\mathbf{v}_\ell^\mathsf{T}\mathbf{x}_2) = y] - (1/2)\right| \geq 1/2r, \quad \forall \mathbf{x}_2 \in \mathcal{X}_2.$$

Since $\mathbf{Ux}$ is a standard Gaussian for any $\mathbf{UU}^\mathsf{T} = \mathbf{I}_m$ and $\|\mathbf{v}_i + \mathbf{v}_j\|_2, \|\mathbf{v}_i - \mathbf{v}_j\|_2 \geq \Omega(\Delta), 1 \leq i < j \leq r$, we have that for $y \in \{\pm 1\}$,

$$\mathbf{E}_{\mathbf{x}\sim\mathcal{N}_n}\left[\left|\sum_{\ell=1}^{r} w_\ell \mathbb{I}[\text{sign}(\mathbf{v}_\ell^\mathsf{T}\mathbf{Ux}) = y] - (1/2)\right|\right]$$

$$\geq \mathbf{Pr}_{\mathbf{x}\sim\mathcal{N}_n}[\mathbf{Ux} \in \mathcal{X}_1] \cdot \mathbf{E}_{\mathbf{x}\sim\mathcal{N}_n}\left[\left|\sum_{\ell=1}^{r} w_\ell \mathbb{I}[\text{sign}(\mathbf{v}_\ell^\mathsf{T}\mathbf{Ux}) = y] - (1/2)\right| \mid \mathbf{Ux} \in \mathcal{X}_1\right]$$

$$+ \mathbf{Pr}_{\mathbf{x}\sim\mathcal{N}_n}[\mathbf{Ux} \in \mathcal{X}_2] \cdot \mathbf{E}_{\mathbf{x}\sim\mathcal{N}_n}\left[\left|\sum_{\ell=1}^{r} w_\ell \mathbb{I}[\text{sign}(\mathbf{v}_\ell^\mathsf{T}\mathbf{Ux}) = y] - (1/2)\right| \mid \mathbf{Ux} \in \mathcal{X}_2\right]$$

$$\geq \Omega(\Delta/r).$$

## C  Omitted Proofs from Section 4

### C.1  Proof of Lemma 4.5

In this section, we prove Lemma 4.5. We start by introducing the following technical results.

**Fact C.1.** *Let $t \geq 2$ and $p, q \in \mathcal{P}_t^d$. Then, we have that*

$$t \int_{\|\mathbf{x}\|_2=1} p(\mathbf{x})q(\mathbf{x})d\mathbf{x} = \frac{1}{d+2t-2}\int_{\|\mathbf{x}\|_2=1}\langle\nabla p(\mathbf{x}), \nabla q(\mathbf{x})\rangle d\mathbf{x} + \frac{1}{d+2t-2}\int_{\|\mathbf{x}\|_2=1} p(\mathbf{x})\nabla^2 q(\mathbf{x})d\mathbf{x}.$$

*Proof of Fact C.1.* Applying the Gaussian Divergence theorem for the function $p(\mathbf{x})\nabla p(\mathbf{x})$ over the unit ball, we have that

$$t \int_{\|\mathbf{x}\|_2=1} p(\mathbf{x})q(\mathbf{x})d\mathbf{x} = \int_{\|\mathbf{x}\|_2=1}\langle p(\mathbf{x})\nabla q(\mathbf{x}), \mathbf{x}\rangle d\mathbf{x} = \int_{\|\mathbf{x}\|_2\leq 1}\nabla\cdot(p(\mathbf{x})\nabla q(\mathbf{x}))d\mathbf{x}$$

$$= \int_{\|\mathbf{x}\|_2\leq 1}\langle\nabla p(\mathbf{x}), \nabla q(\mathbf{x})\rangle d\mathbf{x} + \int_{\|\mathbf{x}\|_2\leq 1} p(\mathbf{x})\nabla^2 q(\mathbf{x})d\mathbf{x}$$

$$= \int_0^1 r^{d-1}dr \int_{\|\mathbf{x}\|_2=1}\langle\nabla p(r\mathbf{x}), \nabla q(r\mathbf{x})\rangle d\mathbf{x} + \int_0^1 r^{d-1}dr \int_{\|\mathbf{x}\|_2=1} p(r\mathbf{x})\nabla^2 q(r\mathbf{x})d\mathbf{x}$$

$$= \int_0^1 r^{2t+d-3}dr \int_{\|\mathbf{x}\|_2=1}\langle\nabla p(\mathbf{x}), \nabla q(\mathbf{x})\rangle d\mathbf{x} + \int_0^1 r^{2t+d-3}dr \int_{\|\mathbf{x}\|_2=1} p(\mathbf{x})\nabla^2 q(\mathbf{x})d\mathbf{x}$$

$$= \frac{1}{d+2t-2}\int_{\|\mathbf{x}\|_2=1}\langle\nabla p(\mathbf{x}), \nabla q(\mathbf{x})\rangle d\mathbf{x} + \frac{1}{d+2t-2}\int_{\|\mathbf{x}\|_2=1} p(\mathbf{x})\nabla^2 q(\mathbf{x})d\mathbf{x} .$$

This completes the proof. $\qquad\square$

**Fact C.2** (see, e.g., Lemma 28 in [Kan15]). *For any $p \in \Omega_t^d$, we have that*

$$\sup_{\|\mathbf{x}\|_2=1} |p(\mathbf{x})| \leq \sqrt{N_{t,d}}\sqrt{\mathbf{E}[p(\mathbf{x})^2]} = \sqrt{N_{t,d}}\|p\|_2.$$

The following lemma provides upper and lower bounds for the expectation of the $L^2$-norm square of the gradient of any homogeneous polynomial $p \in \Omega_t^d$ over the unit sphere $\mathbb{S}^{d-1}$.

**Lemma C.3.** *Let $t$ be an odd positive integer. For any $p \in \mathcal{P}_t^d$, we have that $\mathbf{E}[\|\nabla_o p(\mathbf{x})\|_2^2] \geq (d-1)\|p\|_2^2$ and $\mathbf{E}[\|\nabla p(\mathbf{x})\|_2^2] \leq t(d+2t-2)\|p\|_2^2$.*

*Proof.* By Fact C.1, we have that

$$t(d+2t-2)\|p\|_2^2 = \mathbf{E}[\|\nabla p(\mathbf{x})\|_2^2] + \mathbf{E}[p(\mathbf{x})\nabla^2 p(\mathbf{x})].$$

We bound $\mathbf{E}[p(\mathbf{x})\nabla^2 p(\mathbf{x})]$ as follows. We consider the linear transformations $\mathcal{A}_t : \mathcal{P}_t^d \to \mathcal{P}_{t+2}^d, \mathcal{B}_t :$
$\mathcal{P}_t^d \to \mathcal{P}_{t-2}^d$ as follows: $\mathcal{A}_t(p) = \mathbf{x}^\mathsf{T}\mathbf{x}p(\mathbf{x}), \mathcal{B}_t(p) = \nabla^2 p(\mathbf{x}), p \in \mathcal{P}_t^d$. We first show that for any
$t \geq 2$, both $\mathcal{A}_{t-2}\mathcal{B}_t$ and $\mathcal{B}_{t+2}\mathcal{A}_t$ are symmetric. For any $p, q \in \mathcal{P}_t^d$, applying Fact C.1 yields

$$\langle \mathcal{A}_{t-2}\mathcal{B}_t p, q \rangle = \langle \mathcal{B}_{t+2}\mathcal{A}_t p, q \rangle = \mathbf{E}[\nabla^2 p(\mathbf{x})q(\mathbf{x})]$$
$$= t(d + 2t - 2)\mathbf{E}[p(\mathbf{x})q(\mathbf{x})] - \mathbf{E}[\langle \nabla p(\mathbf{x}), \nabla q(\mathbf{x})\rangle]$$
$$= \mathbf{E}[\nabla^2 q(\mathbf{x})p(\mathbf{x})] = \langle \mathcal{A}_{t-2}\mathcal{B}_t q, p \rangle = \langle \mathcal{B}_{t+2}\mathcal{A}_t q, p \rangle.$$

Therefore, by the eigendecomposition of symmetric linear transformations, we have that $\lambda_1 \|p\|_2^2 \leq$
$\langle \mathcal{A}_{t-2}\mathcal{B}_t p, p \rangle = \mathbf{E}[p(\mathbf{x})\nabla^2 p(\mathbf{x})] \leq \lambda_t \|p\|_2^2, \forall p \in \Omega_t^d$, where $\lambda_1 \leq \cdots \leq \lambda_t$ denote the eigenvalues
of $\mathcal{A}_{t-2}\mathcal{B}_t$. In addition, by elementary calculation, for any $p \in \mathcal{P}_t^d$,

$$\mathcal{B}_{t+2}\mathcal{A}_t p = \nabla^2 \mathbf{x}^\mathsf{T}\mathbf{x}p(\mathbf{x}) = \nabla \cdot (2p(\mathbf{x})\mathbf{x} + \mathbf{x}^\mathsf{T}\mathbf{x}\nabla p(\mathbf{x})) = \sum_{i=1}^d \frac{\partial(2p(\mathbf{x})x_i + \mathbf{x}^\mathsf{T}\mathbf{x}(\nabla p(\mathbf{x}))_i)}{\partial x_i}$$
$$= 2dp(\mathbf{x}) + 4\langle \mathbf{x}, \nabla p(\mathbf{x})\rangle + \mathbf{x}^\mathsf{T}\mathbf{x}\nabla^2 p(\mathbf{x}) = (\mathcal{A}_{t-2}\mathcal{B}_t + 2d + 4t)p.$$

If $\mathcal{A}_{t-2}\mathcal{B}_t$ has an eigenvector $p^*$ corresponding to some eigenvalue $\lambda^*$, then $(\mathcal{A}_t \mathcal{B}_{t+2})(\mathcal{A}_t p^*) =$
$\mathcal{A}_t \mathcal{A}_{t-2}\mathcal{B}_t p^* + (2d + 4t)\mathcal{A}_t p^* = (\lambda^* + 2d + 4t)\mathcal{A}_t p^*$, which implies that $\mathcal{A}_t p^*$ is an eigenvector
of $\mathcal{A}_t \mathcal{B}_{t+2}$ corresponding to the eigenvalue $\lambda^* + 2d + 4t$. Note that since $\mathcal{B}_{t+2}$ maps $\mathcal{P}_{t+2}^d$ to
$\mathcal{P}_t^d$, we have that $\ker(\mathcal{B}_{t+2}) \geq N_{t+2,d} - N_{t,d}$, which implies that $\mathcal{A}_t \mathcal{B}_{t+2}$ has eigenvalue 0 with
multiplicity at least $N_{t+2,d} - N_{t,d}$. Therefore, the eigenvalues of $\mathcal{A}_t \mathcal{B}_{t+2}$ are $0 < \lambda_1 + 2d + 4t \leq$
$\cdots \leq \lambda_t + 2d + 4t$, where the multiplicity of eigenvalue 0 is $N_{t+2,d} - N_{t,d}$ and the multiplicity of
eigenvalue $\lambda_i + 2d + 4t$ is the same as the multiplicity of eigenvalue $\lambda_i$ of $\mathcal{A}_{t-2}\mathcal{B}_t$. Therefore, we
have that $\lambda_1 = 0$ and $\lambda_t = (t - 1)(d + t - 1)$, which implies that

$$\mathbf{E}[\|\nabla p(\mathbf{x})\|_2^2] = t(d + 2t - 2)\|p\|_2^2 - \mathbf{E}[p(\mathbf{x})\nabla^2 p(\mathbf{x})] \in [(t^2 + d - 1)\|p\|_2^2, t(d + 2t - 2)\|p\|_2^2].$$

Therefore, we have that $\mathbf{E}[\|\nabla_o p(\mathbf{x})\|_2^2] = \mathbf{E}[\|\nabla p(\mathbf{x})\|_2^2 - \langle \mathbf{x}, \nabla p(\mathbf{x})\rangle^2] = \mathbf{E}[\|\nabla p(\mathbf{x})\|_2^2] - t^2 \|p\|_2^2 \geq$
$(d - 1)\|p\|_2^2$, completing the proof. $\qquad\square$

We need the following technical lemma which provides a universal upper bound for the $L_2^2$-norm of the gradient of any homogeneous polynomial $p \in \Omega_t^d$.

**Lemma C.4.** *For any $p \in \Omega_t^d$ and any $1 \leq j \leq t$, we have that*

$$\sup_{\|\mathbf{x}\|_2=1} \left\|\frac{\partial^j p(\mathbf{y})}{\partial \mathbf{y}^j}\right\|_2^2 \leq t^j (d + 2t - 2)^j N_{2(t-j),d}\|p\|_2^2.$$

*Proof.* Note that $\|\nabla p(\mathbf{x})\|_2^2 \in \Omega_{2(t-1)}^d$, by Fact C.2, we have that

$$\sup_{\|\mathbf{x}\|_2=1} \|\nabla p(\mathbf{x})\|_2^2 \leq \sqrt{N_{2(t-1),d}}\sqrt{\mathbf{E}[\|\nabla p(\mathbf{x})\|_2^4]} \leq \sqrt{N_{2(t-1),d}}\sqrt{\mathbf{E}[\|\nabla p(\mathbf{x})\|_2^2]}\sqrt{\sup_{\|\mathbf{x}\|_2=1}\|\nabla p(\mathbf{x})\|_2^2},$$

which implies that $\sup_{\|\mathbf{x}\|_2=1}\|\nabla p(\mathbf{x})\|_2^2 \leq N_{2(t-1),d}\mathbf{E}[\|\nabla p(\mathbf{x})\|_2^2] \leq t(d + 2t - 2)N_{2(t-1),d}\|p\|_2^2$.

Since $\left\|\frac{\partial^j p(\mathbf{x})}{\partial \mathbf{x}^j}\right\|_2^2 \leq \left\|\frac{\partial^j p(\mathbf{x})}{\partial \mathbf{x}^j}\right\|_F^2$, it suffices to obtain an upper bound for $\sup_{\|\mathbf{x}\|_2=1}\left\|\frac{\partial^j p(\mathbf{x})}{\partial \mathbf{x}^j}\right\|_F^2$.
Noting that $\left\|\frac{\partial^j p(\mathbf{x})}{\partial \mathbf{x}^j}\right\|_F^2 \in \Omega_{2(t-j)}^d$, by Fact C.2, we have that

$$\sup_{\|\mathbf{x}\|_2=1} \left\|\frac{\partial^j p(\mathbf{x})}{\partial \mathbf{x}^j}\right\|_F^2 \leq \sqrt{N_{2(t-j),d}}\sqrt{\mathbf{E}\left[\left\|\frac{\partial^j p(\mathbf{x})}{\partial \mathbf{x}^j}\right\|_F^4\right]}$$
$$\leq \sqrt{N_{2(t-j),d}}\sqrt{\mathbf{E}\left[\left\|\frac{\partial^j p(\mathbf{x})}{\partial \mathbf{x}^j}\right\|_F^2\right]}\sqrt{\sup_{\|\mathbf{x}\|_2=1}\left\|\frac{\partial^j p(\mathbf{x})}{\partial \mathbf{x}^j}\right\|_F^2},$$

which implies that $\sup_{\mathbf{x} \in \mathbb{S}^{d-1}} \left\| \frac{\partial^j p(\mathbf{x})}{\partial \mathbf{x}^j} \right\|_F^2 \le N_{2(t-j),d} \mathbf{E} \left[ \left\| \frac{\partial^j p(\mathbf{x})}{\partial \mathbf{x}^j} \right\|_F^2 \right]$. Noting that $\frac{\partial p(\mathbf{x})}{\partial x_i} \in \Omega_{t-1}^d$, by Lemma C.3, we have that

$$
\mathbf{E} \left[ \left\| \frac{\partial^2 p(\mathbf{x})}{\partial \mathbf{x}^2} \right\|_F^2 \right] = \mathbf{E} \left[ \sum_{i_1, i_2 \in [d]} \left( \frac{\partial^2 p(\mathbf{x})}{\partial x_{i_1} \partial x_{i_2}} \right)^2 \right] = \sum_{i_1=1}^d \mathbf{E} \left[ \sum_{i_2=1}^d \left( \frac{\partial}{\partial x_{i_2}} \left( \frac{\partial p(\mathbf{x})}{\partial x_{i_1}} \right) \right)^2 \right]
$$

$$
\le (t-1)(d+2t-4) \sum_{i_1=1}^d \mathbf{E} \left[ \left( \frac{\partial p(\mathbf{x})}{\partial x_{i_1}} \right)^2 \right] \le t(d+2t-2) \mathbf{E}[\|\nabla p(\mathbf{x})\|_2^2] \le t^2 (d+2t-2)^2 \|p\|_2^2 .
$$

In general, noting that $\frac{\partial^{j-1} p(\mathbf{x})}{\partial x_{i_1} \cdots \partial x_{i_{j-1}}} \in \Omega_{t-j+1}^d$, by Lemma C.3, we have that

$$
\mathbf{E} \left[ \left\| \frac{\partial^j p(\mathbf{x})}{\partial \mathbf{x}^j} \right\|_F^2 \right] = \mathbf{E} \left[ \sum_{i_1, \ldots, i_j \in [d]} \left( \frac{\partial^2 p(\mathbf{x})}{\partial x_{i_1} \ldots \partial x_{i_j}} \right)^2 \right]
$$

$$
= \sum_{i_1, \ldots, i_{j-1} \in [d]} \mathbf{E} \left[ \sum_{i_j=1}^d \left( \frac{\partial}{\partial x_{i_j}} \left( \frac{\partial^{j-1} p(\mathbf{x})}{\partial x_{i_1} \ldots x_{i_{j-1}}} \right) \right)^2 \right]
$$

$$
\le (t-j+1)(d+2(t-j)) \sum_{i_1, \ldots, i_{j-1} \in [d]} \mathbf{E} \left[ \left( \frac{\partial^{j-1} p(\mathbf{x})}{\partial x_{i_1} \ldots x_{i_{j-1}}} \right)^2 \right]
$$

$$
\le t(d+2t-2) \mathbf{E} \left[ \left\| \frac{\partial^{j-1} p(\mathbf{x})}{\partial \mathbf{x}^{j-1}} \right\|_F^2 \right] \le t^j (d+2t-2)^j \|p\|_2^2 .
$$

Therefore, we have that

$$
\sup_{\|\mathbf{x}\|_2=1} \left\| \frac{\partial^j p(\mathbf{x})}{\partial \mathbf{x}^j} \right\|_F^2 \le N_{2(t-j),d} \mathbf{E} \left[ \left\| \frac{\partial^j p(\mathbf{x})}{\partial \mathbf{x}^j} \right\|_F^2 \right] \le t^j (d+2t-2)^j N_{2(t-j),d} \|p\|_2^2 .
$$

This completes the proof. $\qquad \square$

*Proof of Lemma 4.5.* By definition of $\nabla_o p(\mathbf{y})$, we have that

$$
p(\mathbf{z}) - p(\mathbf{y}) = \frac{p(\mathbf{y} + \delta \cdot \nabla_o p(\mathbf{y}))}{\|\mathbf{y} + \delta \cdot \nabla_o p(\mathbf{y})\|_2^t} - p(\mathbf{y})
$$

$$
= \frac{p(\mathbf{y} + \delta \cdot \nabla_o p(\mathbf{y})) - p(\mathbf{y})}{(1 + \delta^2 \|\nabla_o p(\mathbf{y})\|_2^2)^{t/2}} - \left( 1 - \frac{1}{(1 + \delta^2 \|\nabla_o p(\mathbf{y})\|_2^2)^{t/2}} \right) p(\mathbf{y})
$$

$$
\ge \frac{p(\mathbf{y} + \delta \cdot \nabla_o p(\mathbf{y})) - p(\mathbf{y})}{(1 + \delta^2 \|\nabla_o p(\mathbf{y})\|_2^2)^{t/2}} - \left( 1 - \exp(-t\delta^2 \|\nabla_o p(\mathbf{y})\|_2^2/2) \right) |p(\mathbf{y})|
$$

$$
\ge \frac{p(\mathbf{y} + \delta \cdot \nabla_o p(\mathbf{y})) - p(\mathbf{y})}{(1 + \delta^2 \|\nabla_o p(\mathbf{y})\|_2^2)^{t/2}} - t\delta^2 \|\nabla_o p(\mathbf{y})\|_2^2 |p(\mathbf{y})|/2 .
$$

We bound $p(\mathbf{y} + \delta \cdot \nabla_o p(\mathbf{y})) - p(\mathbf{y})$ as follows: Let $f(s) = p(\mathbf{y} + s\mathbf{v})$ for some unit vector $v \in \mathbb{R}^d$. Noting that $p$ is a degree-$t$ homogeneous polynomial, by Taylor expansion, we have that $f(s) = f(0) + \sum_{j=1}^t \frac{f^{(j)}(0)s^j}{j!}$. By elementary calculation, we have that $f'(0) = \mathbf{v}^\intercal \nabla p(\mathbf{y})$, $f''(0) = \mathbf{v}^\intercal \frac{\partial^2 p(\mathbf{y})}{\partial \mathbf{y}^2} \mathbf{v}, \ldots, f^{(t)}(0) = \left\langle \mathbf{v}^{\otimes t}, \frac{\partial^t p(\mathbf{y})}{\partial \mathbf{y}^t} \right\rangle$. By taking $\mathbf{v}$ to be the direction of $\nabla_o p(\mathbf{y})$, i.e., $\mathbf{v} = \frac{\nabla_o p(\mathbf{y})}{\|\nabla_o p(\mathbf{y})\|_2}$, we have that

$$
p(\mathbf{y} + \delta \cdot \nabla_o p(\mathbf{y})) - p(\mathbf{y}) = f(\delta \|\nabla_o p(\mathbf{y})\|_2) - f(0) = \sum_{j=1}^t \frac{\left\langle \nabla_o p(\mathbf{y})^{\otimes j}, \frac{\partial^j p(\mathbf{y})}{\partial \mathbf{y}^j} \right\rangle \delta^j}{j!} .
$$

Noting that the first order term is $\delta \|\nabla_o p(\mathbf{y})\|_2^2$, it suffices to show that the absolute value of $\sum_{j=2}^t \frac{\left\langle \nabla_o p(\mathbf{y})^{\otimes j}, \frac{\partial^j p(\mathbf{y})}{\partial \mathbf{y}^j} \right\rangle \delta^j}{j!}$ is sufficiently small. Applying Lemma C.4 yields

$$\left| \sum_{j=2}^t \frac{\left\langle \nabla_o p(\mathbf{y})^{\otimes j}, \nabla^j p(\mathbf{y}) \right\rangle \delta^j}{j!} \right| \leq \sum_{j=2}^t \frac{\delta^j \|\nabla_o p(\mathbf{y})\|_2^j \left\| \frac{\partial^j p(\mathbf{y})}{\partial \mathbf{y}^j} \right\|_2}{j!}$$

$$= \delta \|\nabla_o p(\mathbf{y})\|_2^2 \sum_{j=2}^t \frac{\delta^{j-1} \|\nabla_o p(\mathbf{y})\|_2^{j-2} \left\| \frac{\partial^j p(\mathbf{y})}{\partial \mathbf{y}^j} \right\|_2}{j!}$$

$$\leq \delta \|\nabla_o p(\mathbf{y})\|_2^2 \left( \sum_{j=2}^t \frac{\delta^{j-1} \|\nabla p(\mathbf{y})\|_2^{2j-4}}{2j!} + \sum_{j=2}^t \frac{\delta^{j-1} \left\| \frac{\partial^j p(\mathbf{y})}{\partial \mathbf{y}^j} \right\|_2^2}{2j!} \right)$$

$$\leq \delta \|\nabla_o p(\mathbf{y})\|_2^2 \left( \sum_{j=2}^t \frac{\delta^{j-1} (t(d + 2t - 2) N_{2(t-1),d})^{j-2}}{2j!} + \sum_{j=2}^t \frac{\delta^{j-1} t^j (d + 2t - 2)^j N_{2(t-j),d}}{2j!} \right) .$$

Therefore, we will have that $p(\mathbf{y} + \delta \cdot \nabla_o p(\mathbf{y})) - p(\mathbf{y}) \geq C' \delta \|\nabla_o p(\mathbf{y})\|_2^2$ for some universal constant $0 < C' < 1$, as long as $\delta \leq 1/N_{2t,d}^2$. Thus, by Lemma C.3, we have that

$$p(\mathbf{z}) - p(\mathbf{y}) \geq \frac{p(\mathbf{y} + \delta \cdot \nabla_o p(\mathbf{y})) - p(\mathbf{y})}{(1 + \delta^2 \|\nabla_o p(\mathbf{y})\|_2^2)^{t/2}} - t \delta^2 \|\nabla_o p(\mathbf{y})\|_2^2 |p(\mathbf{y})|/2$$

$$= \frac{C' \delta \|\nabla_o p(\mathbf{y})\|_2^2}{(1 + \delta^2 \|\nabla_o p(\mathbf{y})\|_2^2)^{t/2}} - t \delta^2 \|\nabla_o p(\mathbf{y})\|_2^2 |p(\mathbf{y})|/2$$

$$= C' \delta \|\nabla_o p(\mathbf{y})\|_2^2 \exp(-t\delta^2 \|\nabla_o p(\mathbf{y})\|_2^2/2) - t\delta^2 \|\nabla_o p(\mathbf{y})\|_2^2 |p(\mathbf{y})|/2$$

$$\geq C' \delta \|\nabla_o p(\mathbf{y})\|_2^2 \left( 1 - t\delta^2 \|\nabla p(\mathbf{y})\|_2^2/2 - t\delta |p(\mathbf{y})|/2C' \right)$$

$$\geq \delta \|\nabla_o p(\mathbf{y})\|_2^2 \left( C'(1 - t^2\delta^2 (d + 2t - 2) N_{2(t-1),d}/2) - t\delta \sqrt{N_{t,d}}/2 \right)$$

$$\geq C \delta \|\nabla_o p(\mathbf{y})\|_2^2 ,$$

for some universal constant $0 < C < 1$, as long as $\delta \leq 1/N_{2t,d}^2$. This completes the proof. $\square$

## C.2 Proof of Lemma 4.4

Let $p_1, \ldots, p_N \in \Omega$ be an orthonormal basis, i.e., $\mathbf{E}[p_i(\mathbf{x}) p_j(\mathbf{x})] = \mathbb{I}[i = j]$. Let vector $\mathbf{p}(\mathbf{x}) \overset{\text{def}}{=} [p_1(\mathbf{x}), \ldots, p_N(\mathbf{x})]$. We have that $\mathbf{E}[\mathbf{p}(\mathbf{x})] = 0$ and $\mathbf{Cov}[\mathbf{p}(\mathbf{x})] = \mathbf{I}_N$.

$$\mathbf{Pr}\left[ \left\| \frac{1}{r} \sum_{i=1}^r \mathbf{p}(\mathbf{x}_i) \right\|_2 \geq \eta \right] = \mathbf{Pr}\left[ \frac{1}{r^2} \left\| \sum_{i=1}^r \mathbf{p}(\mathbf{x}_i) \right\|_2^2 \geq \eta^2 \right] = \mathbf{Pr}\left[ \frac{1}{r^2} \sum_{j=1}^N \left( \sum_{i=1}^r p_j(\mathbf{x}_i) \right)^2 \geq \eta^2 \right]$$

$$\leq \frac{1}{\eta^2 r^2} \sum_{j=1}^N \mathbf{E}\left[ \left( \sum_{i=1}^r p_j(\mathbf{x}_i) \right)^2 \right] = \frac{N}{r\eta^2}.$$

We now assume that $\frac{1}{r} \left\| \sum_{i=1}^r \mathbf{p}(\mathbf{x}_i) \right\|_2 \leq \eta$. Let $p \in \Omega$ be an arbitrary polynomial. We can write $p(\mathbf{x}) = \sum_{j=1}^N \alpha_j p_j(\mathbf{x})$, where $\|p\|_2^2 = \sum_{j=1}^N \alpha_j^2$. We have that

$$\frac{1}{r} \left| \sum_{i=1}^r p(\mathbf{x}_i) \right| = \frac{1}{r} \left| \sum_{i=1}^r \sum_{j=1}^N \alpha_j p_j(\mathbf{x}_i) \right| \leq \frac{1}{r} \sum_{j=1}^N |\alpha_j| \left| \sum_{i=1}^r p_j(\mathbf{x}_i) \right|$$

$$\leq \frac{1}{r} \sqrt{\sum_{j=1}^N \alpha_j^2} \sqrt{\sum_{j=1}^N \left( \sum_{i=1}^r p_j(\mathbf{x}_i) \right)^2} = \frac{\|p\|_2}{r} \left\| \sum_{i=1}^r \mathbf{p}(\mathbf{x}_i) \right\|_2 \leq \eta \|p\|_2,$$

where the second inequality follows from Cauchy-Schwarz. This completes the proof.

## C.3 Proof of Theorem 4.6

Let $p \in \partial \Omega_t^d$. Since

$$\|\nabla p(\mathbf{y})\|_2^2 = \|\nabla_o p(\mathbf{y})\|_2^2 + \langle \mathbf{y}, \nabla p(\mathbf{y}) \rangle^2 = \|\nabla_o p(\mathbf{y})\|_2^2 + t^2 p(\mathbf{y})^2 ,$$

by Lemma 4.5, we have that $p(\mathbf{z}) \geq p(\mathbf{y}) + C\delta(\|\nabla p(\mathbf{y})\|_2^2 - t^2 p(\mathbf{y})^2)$. Let

$$q(\mathbf{y}) = p(\mathbf{y}) + C\delta\|\nabla_o p(\mathbf{y})\|_2^2 = p(\mathbf{y}) + C\delta(\|\nabla p(\mathbf{y})\|_2^2 - t^2 p(\mathbf{y})^2) .$$

By definition, we have that $q(\mathbf{y}) - \mathbf{E}[q(\mathbf{y})] = p(\mathbf{y}) + C\delta(\|\nabla p(\mathbf{y})\|_2^2 - t^2 p(\mathbf{y})^2) - C\delta\mathbf{E}[\|\nabla_o p(\mathbf{y})\|_2^2]$, which is a polynomial of degree at most $2t$ and contains only monomials of degree $2t, 2t-2, t, 0$. Let $\Omega$ be the subspace of polynomials in $d$-variables containing all monomials of degree $2t, 2t - 2, t, 0$. In this way, the dimension of $\Omega$ is

$$N = \binom{d + 2t - 1}{d - 1} + \binom{d + 2t - 3}{d - 1} + \binom{d + t - 1}{d - 1} + 1 \leq 3N_{2t,d} .$$

Applying Lemma 4.4 yields that with probability at least $1 - \frac{N}{r\eta^2}$, we have that $\left|\frac{1}{r}\sum_{i=1}^r q(\mathbf{y}_i) - \mathbf{E}[q(\mathbf{y})]\right| \leq \eta\|q(\mathbf{y}) - \mathbf{E}[q(\mathbf{y})]\|_2, \forall q \in \Omega$. Therefore, we have that

$$\frac{1}{r}\sum_{i=1}^r p(\mathbf{z}_i) \geq \frac{1}{r}\sum_{i=1}^r q(\mathbf{y}_i) \geq \mathbf{E}[q(\mathbf{y})] - \eta\|q(\mathbf{y}) - \mathbf{E}[q(\mathbf{y})]\|_2 = \mathbf{E}[q(\mathbf{y})] - \eta\sqrt{\mathbf{E}[q(\mathbf{y})^2] - \mathbf{E}[q(\mathbf{y})]^2}.$$

By elementary calculation, we have that

$$\begin{aligned}
\mathbf{E}[q(\mathbf{y})^2] - \mathbf{E}[q(\mathbf{y})]^2 &= \mathbf{E}[(p(\mathbf{y}) + C\delta\|\nabla_o p(\mathbf{y})\|_2^2)^2] - C^2\delta^2\mathbf{E}[\|\nabla_o p(\mathbf{y})\|_2^2]^2 \\
&= \mathbf{E}[p(\mathbf{y})^2] + 2C\delta\mathbf{E}[p(\mathbf{y})\|\nabla_o p(\mathbf{y})\|_2^2] + C^2\delta^2\mathbf{E}[\|\nabla_o p(\mathbf{y})\|_2^4] - C^2\delta^2\mathbf{E}[\|\nabla_o p(\mathbf{y})\|_2^2]^2 \\
&= \mathbf{E}[p(\mathbf{y})^2] + C^2\delta^2\mathbf{E}[\|\nabla_o p(\mathbf{y})\|_2^4] - C^2\delta^2\mathbf{E}[\|\nabla_o p(\mathbf{y})\|_2^2]^2 \\
&= 1 + C^2\delta^2\mathbf{E}[\|\nabla_o p(\mathbf{y})\|_2^4] - C^2\delta^2\mathbf{E}[\|\nabla_o p(\mathbf{y})\|_2^2]^2,
\end{aligned}$$

where the second equality is due to $p(\mathbf{y})$ being odd and

$$\begin{aligned}
\|\nabla_o p(-\mathbf{y})\|_2^2 &= \|\nabla p(-\mathbf{y})\|_2^2 - t^2 p(-\mathbf{y})^2 = \|\nabla(-p(\mathbf{y}))\|_2^2 - t^2(-p(\mathbf{y}))^2 \\
&= \|\nabla p(\mathbf{y})\|_2^2 - t^2 p(\mathbf{y})^2 = \|\nabla_o p(\mathbf{y})\|_2^2 .
\end{aligned}$$

By Lemma C.3 and Lemma C.4, we have that $\mathbf{E}[\|\nabla_o p(\mathbf{y})\|_2^2] \geq d - 1$ and $\mathbf{E}[\|\nabla_o p(\mathbf{y})\|_2^4] \leq \mathbf{E}[\|\nabla p(\mathbf{y})\|_2^2]\sup_{\|\mathbf{y}\|_2=1}\|\nabla p(\mathbf{y})\|_2^2 \leq t^2(d + 2t - 2)^2 N_{2(t-1),d}$. Therefore, we have that

$$\begin{aligned}
\frac{1}{r}\sum_{i=1}^r p(\mathbf{z}_i) &\geq \mathbf{E}[q(\mathbf{y})] - \eta\sqrt{\mathbf{E}[q(\mathbf{y})^2] - \mathbf{E}[q(\mathbf{y})]^2} \\
&\geq C\delta\mathbf{E}[\|\nabla_o p(\mathbf{y})\|_2^2] - \eta\sqrt{1 + C^2\delta^2\mathbf{E}[\|\nabla_o p(\mathbf{y})\|_2^4] - C^2\delta^2\mathbf{E}[\|\nabla_o p(\mathbf{y})\|_2^2]^2} \\
&\geq C\delta(d - 1) - \eta\sqrt{1 + C^2\delta^2(t^2(d + 2t - 2)^2 N_{2(t-1),d} - (d - 1)^2)} \\
&= C\delta\left(d - 1 - \eta\sqrt{\frac{1}{C^2\delta^2} + t^2(d + 2t - 2)^2 N_{2(t-1),d} - (d - 1)^2}\right) .
\end{aligned}$$

Taking $\delta = 1/N_{2t,d}^2$ and $\eta = \frac{Cd}{3N_{2t,d}^2}$ yields that with probability at least

$$1 - \frac{N}{r\eta^2} \geq 1 - \frac{27}{C^2 d^2} \geq 99/100 ,$$

$$\begin{aligned}
\frac{1}{r}\sum_{i=1}^r p(\mathbf{z}_i) &\geq C\delta\left(d - 1 - \eta\sqrt{N_{2t,d}^4/C^2 + t^2(d + 2t - 2)^2 N_{2(t-1),d} - (d - 1)^2}\right) \\
&> C\delta\left(d/2 - \eta\sqrt{2N_{2t,d}^4/C^2}\right) \geq 0 .
\end{aligned}$$

## C.4 Omitted Calculations in Proof of Theorem 4.2

By elementary calculation, we have that

$$
\begin{aligned}
\|\mathbf{z}_i^* - \mathbf{y}_i\|_2 &= \left\| \frac{\mathbf{y}_i + \delta\nabla_o p^*(\mathbf{y}_i)}{\|\mathbf{y}_i + \delta\nabla_o p^*(\mathbf{y}_i)\|_2} - \mathbf{y}_i \right\|_2 = \frac{\|\mathbf{y}_i + \delta\nabla_o p^*(\mathbf{y}_i) - \|\mathbf{y}_i + \delta\nabla_o p^*(\mathbf{y}_i)\|_2 \mathbf{y}_i\|_2}{\|\mathbf{y}_i + \delta\nabla_o p^*(\mathbf{y}_i)\|_2} \\
&\leq \frac{|1 - \|\mathbf{y}_i + \delta\nabla_o p^*(\mathbf{y}_i)\|_2| + \delta\|\nabla p^*(\mathbf{y}_i)\|_2}{1 - \delta\|\nabla p^*(\mathbf{y}_i)\|_2} \leq \frac{2\delta\|\nabla p^*(\mathbf{y}_i)\|_2}{1 - \delta\|\nabla p^*(\mathbf{y}_i)\|_2} \leq O(1/N_{2t,d}) \ ,
\end{aligned}
$$

where the last inequality follows from for any $\mathbf{y} \in \mathbb{S}^{d-1}$, $\|\nabla p^*(\mathbf{y})\|_2 \leq \sqrt{t(d + 2t - 2)N_{2(t-1),d}\|p^*\|_2^2} \leq N_{2t,d}$ by Lemma C.3.

## C.5 Omitted Calculations in Proof of Theorem 1.2

In this section, we provide calculation details to show that $r \geq N_{2k,m}$ and $N_{2k,m} \leq \Omega((1/\Delta)^{1.89})$. We have the following chain of inequalities:

$$
N_{2k,m}^5 \leq \binom{m + 2k}{2k}^5 = \binom{(1 + 2c')m}{m}^5 \leq 2^{5(1+2c')mH\left(\frac{1}{1+2c'}\right)} = 2^{5(1+2c')m\left(\frac{\log(1+2c')}{1+2c'} + \frac{2c'\log(1+1/2c')}{1+2c'}\right)}
$$

$$
= 2^{5m\left(\log(1+2c') + 2c'\log(1+1/2c')\right)} \leq 2^{\frac{5c\log r(\log(1+2c') + 2c'\log(1+1/2c'))}{\log(1/\Delta)}} \leq r,
$$

where $H(p) = -p\log p - (1-p)\log(1-p)$, $p \in [0,1]$, is the standard binary entropy function. On the other hand, by our choice of $m$, we have that

$$
N_{2k,m} = \binom{m + 2k - 1}{m - 1} = \binom{(1 + 2c')m - 1}{m - 1} \geq \left(\frac{(1 + 2c')m - 1}{m - 1}\right)^{m-1} \geq (1 + 2c')^{m-1}
$$

$$
\geq (1 + 2c')^{\frac{1.99\log r}{\log(1/\Delta)} - 1} = \left((1/e)(1/\Delta)^{1/5c}\right)^{\frac{1.89\log r}{\log(1/\Delta)}} \geq \Omega((1/\Delta)^{1.89}) \ .
$$