# OpenReview forum: "SQ Lower Bounds for Learning Mixtures of Linear Classifiers"
_NeurIPS.cc/2023/Conference — NeurIPS 2023 poster_

### Official Review · Reviewer_n8Pi · 2023-07-04

**Soundness:** 3 good
**Presentation:** 4 excellent
**Contribution:** 2 fair
**Rating:** 5
**Confidence:** 3

**Summary:**

This paper studies the problem of learning mixtures of linear classifiers under Gaussian sampling. The paper provides a stastical query lower bound which demonstrates that known algorithms for the problem in the literature are essentially best possible, even for the special case of learning uniform mixtures. It further establishes the complexity of any SQ algorithm.

**Strengths:**

+) Statistical query complexity of learning mixtures of linear classifiers
+) Efficient spherical designs to fullfill the required separation assumptions for the results to hold

**Weaknesses:**

This paper is primarily a theoretical work. The assumptions, problem setup, and results are only of theoretical interest. I am not sure if the results and technical tools are interesting to the broad machine learning systems society.

**Questions:**

The paper is well written and easy to follow. There are some notation issues. For example, in the abstract, "y=sign(<v_\ell,x>))" where y shall be y_\ell right? This also appeared in several places in the paper. Plus, one right parenthesis ) shall be removed.

---

> ### Author Rebuttal · Authors · 2023-08-09
>
> We thank the reviewer for their feedback and effort. We will address the typos pointed out in the revision.
>
> The reviewer’s stated weakness of our work is the fact that it is “primarily a theoretical work”.
> We respectfully point out that theoretical research in machine learning (“learning theory”) is well within the scope of NeurIPS’23 and explicitly mentioned in the call for papers. We request that our submission is judged on its merits according to the specified criteria in the call for papers.
>
> Specifically, the problem that we study (learning mixtures of linear classifiers) is a classical problem in machine learning and understanding its computational complexity is a question of fundamental importance. Our work provides near-optimal SQ lower bounds for this problem, suggesting that known algorithms are essentially best possible. In the process, our work develops novel constructions of spherical designs that are of independent interest. In summary, we believe that our contributions are of significant interest to the theoretical ML community that has a strong presence at NeurIPS.

---

> > ### Comment · Reviewer_n8Pi · 2023-08-21
> > **Thanks for the rebuttal!**
> >
> > Thanks the authors for the rebuttal and for addressing the issues. After reading the rebuttal and other reviewers' comments, I am updating thescore to Borderline accept.

---

### Official Review · Reviewer_Azis · 2023-07-04

**Soundness:** 4 excellent
**Presentation:** 3 good
**Contribution:** 3 good
**Rating:** 7
**Confidence:** 3

**Summary:**

A statistical query (SQ) algorithm is an algorithm that attempts to learn the data distribution $D$ by querying $f$ to the oracle who responses with $v$ such that $\lvert v - E_{x\sim D} f(x) \rvert$ is small. In this paper, the authors study the problem of finding an SQ lower bound for learning a mixture of linear classifiers. The main ingredients are

1. a result in [DKPZ21], which establishes an SQ lower bound on testing a distribution of $(x,y)\in \mathbb{R}^n\times \{-1,1\}$ where $x$ is normal, $E[y \vert x = z]=g(Uz)$ for some function $g$ with zero low-degree moments and some matrix $U$.
2. Authors' construction of well-separated _spherical design_, that is, unit vectors $v_1,\ldots,v_r$ such that $g(z) = \frac1{r}\sum_{\ell=1}^r\text{sign} (v_{\ell}^Tz)$ has zero low-degree moments, as required in 1.

And the last step is to turn the testing problem into the problem of learning the mixture of linear classifiers. The proof of the main theorem is presented in the main paper, whereas those of several lemmas are postponed to the appendix.



**Strengths:**

- The authors have established a new SQ lower bound which matches the algorithmic guarantees in some cases [CDV22].
- The idea of designing and using spherical designs to obtain the lower bound is interesting and can be applied to other learning problems.
- The writing is well-organized and it is easy to see the high-level idea of the proof.

**Weaknesses:**

My only minor concern is that there is a lack of discussion of the main result. See Questions below for some questions that I have in mind.



**Questions:**

1. the authors have mentioned that the main result provides "a near-optimal information-computation tradeoff for the problem". Does this mean that the problem of finding an optimal tradeoff is still open.
2. Related to 1., are there any open problems introduced, or related to this work?
3. The idea of using spherical designs could be used to find SQ lower bound of other learning problems in which $g$ is an odd function. Are there any other learning problems that the authors have in mind?
4. Can the result be extended to non-normal data?

**Limitations:**

The authors should include a section on discussion, or limitations of the main results. For example, I have already mentioned the case where the data is non-normal.

Also, am I correct in assuming that the result only applies when $r$ is already known. Is there anything we can do when $r$ is not known?

---

> ### Author Rebuttal · Authors · 2023-08-09
>
> We would like to thank the reviewer for their effort and positive assessment of our work.
>
> We start by addressing the reviewer’s concerns in the “Limitations” sections below:
>
> 1. “Also, am I correct in assuming that the result only applies when $r$ is already known. Is there anything we can do when $r$ is not known?”
>
> In our work, we prove a lower bound for known $r$, which automatically establishes the hardness for the (more challenging) setting where $r$ is unknown. On the other hand, we do not know any algorithmic result for unknown $r$.
>
>
> 2. “The authors should include a section on discussion, or limitations of the main results.”
>
> After the statement of our main Theorem 1.2, we include a section (line 89 - line 105) to discuss the implications of our result. Our understanding is that a “Limitations” section is not required at NeurIPS’23. If this is not accurate, we will be happy to include one in the revised version.
>
> We now proceed by addressing the reviewer’s questions below:
>
> 1. The authors have mentioned that the main result provides "a near-optimal information-computation tradeoff for the problem". Does this mean that the problem of finding an optimal tradeoff is still open.
>
> Our SQ lower bound qualitatively matches the best known algorithm in the sense that both the upper and the lower bound are of the form $n^{\mathrm{poly}(1/\Delta)\log(r)}$. On the other hand, the degree of the polynomials in the exponent do not exactly match. Specifically, in our SQ lower bound, the exponent on $(1/\Delta)$ is around $1/10$, which is strictly smaller than the one for the algorithmic result of [CDV22].
>
> 2. Are there any open problems introduced, or related to this work?
>
> An interesting research direction is to understand if our techniques can be leveraged to obtain SQ lower bounds for other mixture models (e.g., for other mixtures of experts). In addition to SQ lower bounds, it would also be interesting to establish reduction-based hardness for such problems, starting, e.g., from cryptographic assumptions. A more concrete open problem is to obtain sharper lower bounds for this particular problem (matching the constant in the exponent as well).
>
> 3. The idea of using spherical designs could be used to find SQ lower bound of other learning problems in which $g$ is an odd function. Are there any other learning problems that the authors have in mind?
>
> While the focus of our work has been on learning mixtures of linear classifiers, we believe that a similar approach ought to apply for other “mixtures of experts” problems. We leave this as a direction for future work.
>
>
> 4. Can the result be extended to non-normal data?
>
> The main point of our work is that we establish a hardness result (SQ lower bound), even for the arguably simplest (and well-studied) setting where the covariates are drawn from the standard Gaussian distribution. This implies similar SQ lower bounds, e.g., when the covariates are drawn from a more general distribution family (e.g., an unknown subgaussian or log-concave distribution) that includes the standard normal. If one wants to establish SQ lower bounds for a non-Gaussian fixed distribution on covariates, a different construction is needed.
>
> References:
>
> [CDV22] A. Chen, A. De, and A. Vijayaraghavan. Algorithms for learning a mixture of linear 416 classifiers. In International Conference on Algorithmic Learning Theory, pages 205-226. PMLR, 2022.

---

> > ### Comment · Reviewer_Azis · 2023-08-17
> > **Response**
> >
> > I am satisfied with the authors' answers. A comment:
> >
> > > Our understanding is that a “Limitations” section is not required at NeurIPS’23. If this is not accurate, we will be happy to include one in the revised version.
> >
> > My suggestion is that the authors can add their rebuttal to a "future direction" section; for example:
> >
> > > While the focus of our work has been on learning mixtures of linear classifiers, we believe that a similar approach ought to apply for other “mixtures of experts” problems. We leave this as a direction for future work.
> >
> > > If one wants to establish SQ lower bounds for a non-Gaussian fixed distribution on covariates, a different construction is needed.

---

### Official Review · Reviewer_vDTe · 2023-07-05

**Soundness:** 3 good
**Presentation:** 3 good
**Contribution:** 3 good
**Rating:** 7
**Confidence:** 3

**Summary:**

The paper provides statistical query lower bounds for learning mixture of linear classifier.

In the problem of learning mixture of linear classifier, there are $r$ linear classifier $v_1, \ldots ,v_r \in \mathbb{R}^{n}$. The input feature $x\in \mathbb{R}^{n}$ is draw from gaussian, the label $y = \mathsf{sign}(v_\ell^\top x)$, where the index $\ell$ is chosen with probability $w_{\ell}$ (one has $w_1 +\cdots +w_{r} = 1$). Previous work [CDV'22] give an algorithm of sample complexity $n^{\log(r)/\Delta^2}$ where $\delta$ is the minimum separation between classifiers. The major contribution of this paper is to give an almost matching lower bound of $n^{\log (r) \cdot \mathsf{poly}(\Delta^{-1})}$, under the SQ model.

From a high level, the paper follows the framework of [DKS'17] and reduces the problem to sphere design. The major technical contribution of this paper is to provide a spherical design and find a set of vector from unit ball that satisfies (1) none trivial pairwise distance and (2) the correlation with any low degree polynomial equal $0$. They obtain the spherical design using a topological argument inspired from [BRV'13].


----------------------
I have read the rebuttal and I would keep my positive evaluation of the paper.

**Strengths:**

The paper gives an almost matching SQ lower bound for learning mixture linear classifier, the technique is novel and could have broad applications.

**Weaknesses:**

There is no major weakness, though I have a few minor questions (to be specified later)

**Questions:**

(1) I have some concern with Lemma 3.5. In the statement, it is only required that $f \in L^2(R, N)$, but it seems not true for every such function, right? If I understand correctly, the proof only requires $f = sign$ (i.e., only need to work for sign function), but even for sign function, the claim is sloppy, because $E_{z\sim N_m} [p(z)f(v^\top z)]$ scaling invariant with $v$, which seems not true for polynomial?
Please clarify this point.


(2) Is the algorithm of [CDV'22] falls into the SQ framework?

---

> ### Author Rebuttal · Authors · 2023-08-09
>
> We would like to thank the reviewer for their effort and positive assessment of our work.
>
> We respond to the reviewer’s questions below:
>
> 1. I have some concern with Lemma 3.5. In the statement, it is only required that $f\in L^2(\mathbb{R},\mathcal{N})$, but it seems not true for every such function, right? If I understand correctly, the proof only requires $f = \mathrm{sign}$  (i.e., only need to work for $\mathrm{sign}$ function), but even for $\mathrm{sign}$ function, the claim is sloppy, because $\mathbf{E}_{z\sim \mathcal{N}_m}[p(\mathbf{z})f(\mathbf{v}^\intercal\mathbf{z})]$ scaling invariant with $\mathbf{v}$, which seems not true for polynomial? Please clarify this point.
>
> We thank the reviewer for pointing this out. The statement of Lemma 3.5 requires the assumption that $\mathbf{v}$ is a unit vector (or, more generally, a non-zero vector of fixed $L_2$-norm).
> Please note that we only invoke this lemma for $\mathbf{v}$ being a unit vector. Without this assumption, the conclusion is not true (as the reviewer pointed out). Under the assumption that $\mathbf{v}$ is a unit vector, Lemma 3.5 holds as stated, i.e., for any function $f$ in the space $L^2(\mathbb{R},\mathcal{N})$ (not only for the $\mathrm{sign}$ function).
>
>
> Looking at the proof of Lemma 3.5 in the appendix, we also want to point out that Claim B.1 requires the assumption that both $\mathbf{U}$ and $\mathbf{V}$ are projection matrices (i.e., $\mathbf{U}\mathbf{U}^{\intercal}=I_{n_1}$, and $\mathbf{V}\mathbf{V}^\intercal=I_{n_2}$), because the equations in line 576 hold only if both $\mathbf{U}$ and $\mathbf{V}$ are projection matrices.
>
> 2. Is the algorithm of [CDV22] falls into the SQ framework?
>
> The algorithm in [CDV22] can be implemented in the Statistical Query(SQ) model efficiently. In fact, the class of SQ algorithms is rather broad and captures a range of known supervised learning algorithms. More broadly speaking, several known algorithmic techniques in machine learning are known to be efficiently implementable using SQ algorithms. These include spectral techniques, moment and tensor methods, local search (e.g., Expectation Maximization), and many others (see, e.g., [FGR+17, FGV17]).
>
> References:
>
> [CDV22] A. Chen, A. De, and A. Vijayaraghavan. Algorithms for learning a mixture of linear 416 classifiers. In International Conference on Algorithmic Learning Theory, pages 205-226. PMLR, 2022.
>
> [FGR+17] V. Feldman, E. Grigorescu, L. Reyzin, S. Vempala, and Y. Xiao. Statistical algorithms and a lower bound for detecting planted cliques. J. ACM, 64(2):8:1-8:37, 2017.
>
> [FGV17] V. Feldman, C. Guzman, and S. S. Vempala. Statistical query algorithms for mean vector estimation and stochastic convex optimization. In Proceedings of the Twenty Eighth Annual ACM-SIAM Symposium on Discrete Algorithms, SODA 2017, pages 1265-1277. SIAM, 2017.

---

> > ### Comment · Reviewer_vDTe · 2023-08-12
> > **Thanks for clarification.**
> >
> > Thanks for the clarification. Please consider adding the assumptions on $v$ to the theorem statement.

---

### Official Review · Reviewer_Jk98 · 2023-07-07

**Soundness:** 4 excellent
**Presentation:** 4 excellent
**Contribution:** 3 good
**Rating:** 6
**Confidence:** 2

**Summary:**

The authors prove a lower bound for the number of queries needed in the statistical query model for learning a mixture of linear classifiers. The statistical query model essentially makes oracle queries with a polynomial f(x) and the oracle responds with a value v such that: |v-E[f(x)]\le t, for some threshold t (the accuracy of the oracle). The main result in the paper is Theorem 1.2 which roughly states that to learn a n-dimensional mixture of linear classifiers within TV distance  \eps any algorithm must use queries with accuracy 1/poly(n) or must make 2^poly(n) statistical queries.

**Strengths:**

- The lower bounds are claimed to be tight.
- The problem of learning a mixture of linear classifiers is simple and elegant and thus important from a learning theory viewpoint.
- The technical results in the paper leading to Thm 4.2 synthesize ideas from topology and analysis. From a technical perspective the proof is indeed interesting and informative.

**Weaknesses:**

- The authors claim the lower bound is "qualitatively match" previous results by Chen et al. 2022. It would be good to reduce the Che et al result to the SQ model or vice versa. Otherwise the lower bound loses some of its significance.

**Questions:**

In Theorem 1.2, is it possible to have \Delta not depend upon r i.e., can the dependence \Delta > 1/r^10 be eliminated?

---

> ### Author Rebuttal · Authors · 2023-08-09
>
> We would like to thank the reviewer for their effort and positive assessment of our work.
>
> 1. Regarding the reviewer’s point:
> “The authors claim the lower bound is "qualitatively match" previous results by Chen et al. 2022. It would be good to reduce the Che et al result to the SQ model or vice versa. Otherwise the lower bound loses some of its significance.”
>
> The algorithm in [CDV22] can be implemented in the Statistical Query(SQ) model efficiently. In fact, the class of SQ algorithms is rather broad and captures a range of known supervised learning algorithms. More broadly speaking, several known algorithmic techniques in machine learning are known to be efficiently implementable using SQ algorithms. These include spectral techniques, moment and tensor methods, local search (e.g., Expectation Maximization), and many others (see, e.g., [FGR+17, FGV17]).
>
> We respond to the reviewer’s question below:
>
> 1. In Theorem 1.2, is it possible to have $\Delta$ not depend upon $r$ i.e., can the dependence $\Delta \ge r^{-1/10}$ be eliminated?
>
>
> Our lower bound proof requires the assumption that $\Delta\ge r^{-c}$ for some absolute constant $0<c<1$. We note that this parameter setting is arguably the most interesting in practical settings.
> In our technical proof, the lower bound comes from Proposition 3.2 [DKPZ21], where it requires the low dimension $m \ge2$. In particular, to apply the techniques in our setting, we randomly sample $r$ $\Delta$-separated unit vectors over the $m$-dimensional unit sphere with sufficiently small error guarantees. This is impossible without the dependence on $\Delta$. In addition, by taking $\Delta=r^{-c}$, we do provide a lower bound for small $\Delta$. However, since the algorithmic result of [CDV22] has sample and runtime complexity $\min(n^{O(\log r/\Delta^2)},(n/\Delta)^{O(r)})$, which will be $(n/\Delta)^{O(r)}$ if $\Delta$ is sufficiently small.
> This provides strong evidence that we are not able to obtain near-optimal hardness results without the assumption on $\Delta$.
>
> References:
>
> [CDV22] A. Chen, A. De, and A. Vijayaraghavan. Algorithms for learning a mixture of linear 416 classifiers. In International Conference on Algorithmic Learning Theory, pages 205-226. PMLR, 2022.
>
> [DKPZ] I. Diakonikolas, D. M. Kane, T. Pittas, and N. Zarifis. The optimality of polynomial re435 gression for agnostic learning under gaussian marginals in the SQ model. In Conference 436 on Learning Theory, COLT 2021, volume 134 of Proceedings of Machine Learning Re437 search, pages 1552-1584. PMLR, 2021.
>
> [FGR+17] V. Feldman, E. Grigorescu, L. Reyzin, S. Vempala, and Y. Xiao. Statistical algorithms and a lower bound for detecting planted cliques. J. ACM, 64(2):8:1-8:37, 2017.
>
> [FGV17] V. Feldman, C. Guzman, and S. S. Vempala. Statistical query algorithms for mean vector estimation and stochastic convex optimization. In Proceedings of the Twenty Eighth Annual ACM-SIAM Symposium on Discrete Algorithms, SODA 2017, pages 1265-1277. SIAM, 2017.

---

### Official Review · Reviewer_TFYT · 2023-07-11

**Soundness:** 3 good
**Presentation:** 2 fair
**Contribution:** 3 good
**Rating:** 5
**Confidence:** 2

**Summary:**

The authors study the problem of learning mixture of linear classifiers with Gaussian covariates. Their primary result is a near-optimal SQ lower bound which applies even for the uniform mixture case. Moreover, as a purely mathematical result, they construct an efficient spherical design (under a stronger definition of the structure) with sample complexity within a polylogarithmic factor of the optimum.

**Strengths:**

The paper is well motivated and introduced succinctly. The strongest portion of the submission are the first two sections which nicely introduce the problem, its inherent difficulties, and a discussion of the papers approach. I thought the technical overview was especially helpful to highlight the problem and challenges in proving the major results.

**Weaknesses:**

The paper predominantly lacks in presentation of the mathematical results. Namely the major theoretical results presented in the main text are somewhat obtuse and hard to parse / verify the theoretical claims. This may be due my unfamiliarity with the surrounding literature, however the second half of the paper is difficult to follow in the logic of the proofs. While the technical overview helps to introduce the complexities of the analysis, these final sections are entirely non-intuitive. I would suggest to the authors to reduce the number of theorem / lemma statements in the main text and instead more carefully chose results (in line with the major points of Section 1.2). A more expansion on these results with more fleshed out proofs (or proof sketches) would make the paper considerably more readable within the short page limit of the conference. Additionally, it is somewhat hard to tell where the current result fits in with the prior works on the topic.

**Questions:**

What is the relation of the given problem to that of sparse recovery?

Does the spherical design technique used to prove your main result extend well beyond simple linear classifiers?

How can the novel techniques discussed here be applied more broadly outside of the presented problem instance?

---

> ### Author Rebuttal · Authors · 2023-08-09
>
> We thank the reviewer for their feedback and effort.
> We start by addressing the concerns within the review in order:
>
> 1. Mathematical Contributions and Presentation.
>
> Our main technical contribution is a novel construction of a spherical $t$-design, which leads to a nearly-optimal SQ lower bound for learning uniform mixtures of linear classifiers (see lines 159 and 160). To achieve this, we leverage ideas and results (Theorem 4.3) from the pure mathematics literature [BRV13]. Although the original theorem in [BRV13] is sophisticated and perhaps non-intuitive for a non-expert, we have distilled and simplified the statement so that it can resonate with ML researchers with theoretical background. In order to help the reader follow the proof idea of our spherical $t$-design construction (Theorem 1.5), we provide high-level explanations in the technical review session (Section 1.2), and also break down our technical results in Section 4 into several short lemmas (see Lemmas 4.4, 4.5 and Theorem 4.6), followed by intuitive prose and proof sketches. We would welcome additional concrete suggestions by the reviewer.
>
> 2. Regarding the reviewer’s point:
> “Additionally, it is somewhat hard to tell where the current result fits in with the prior works on the topic.”
>
> In addition to the mathematical topic of spherical designs, the problem of learning mixtures of linear classifiers is important and fundamental from an ML theory viewpoint. In lines 23 and 24, we clearly state the related algorithmic results for this problem. Previous work [CDV22] gave an algorithm with sample and computational complexity $n^{O(\log r/\Delta^2)}$ for the problem, where $\Delta$ is the minimum separation between linear classifiers. In our work, we provide a nearly optimal SQ lower bound of $n^{\mathrm{poly}(1/\Delta)\log r}$.
>
>
> We now proceed by addressing the reviewer’s questions below:
>
> 1. What is the relation of the given problem to that of sparse recovery?
>
> The problem we study (learning mixtures of linear classifiers) and the associated techniques in our work are orthogonal to the problem of sparse recovery.
>
> 2. Does the spherical design technique used to prove your main result extend well beyond simple linear classifiers?
>
> Yes. Our technique works for any odd function in the space $L^2(\mathbb{R},\mathcal{N})$, not only for linear classifiers (corresponding to the $\mathrm{sign}$ function).
>
> 3. How can the novel techniques discussed here be applied more broadly outside of the presented problem instance?
>
> We believe that our new efficient construction of spherical $t$-designs itself is a mathematical contribution of independent interest that could be used to establish SQ lower bounds for other related mixture models (e.g., for mixtures of experts). That said, the focus of our work has been the fundamental problem of learning mixtures of linear classifiers.
>
> References:
>
> [BRV13] A. Bondarenko, D. Radchenko, and M. Viazovska. Optimal asymptotic bounds for spherical designs. Annals of mathematics, pages 443-452, 2013.
>
> [CDV22] A. Chen, A. De, and A. Vijayaraghavan. Algorithms for learning a mixture of linear 416 classifiers. In International Conference on Algorithmic Learning Theory, pages 205-226. PMLR, 2022.

---

> > ### Comment · Reviewer_TFYT · 2023-08-16
> >
> > I thank the reviewer for their response to my questions and comments. I am happy to increase my score to a 5 but highlight to the AC/PC that this work is somewhat outside of my area of expertise so encourage them to consider the other reviews more heavily than my own.

---

### Author Rebuttal · Authors · 2023-08-10

We thank the reviewers for their time and effort in providing feedback. We are encouraged by the positive comments from reviewers (**Jk98**,**vDTe**,**Azis**) for the following:
(i) the importance of the problem we study (learning mixtures of linear classifiers)
and the tightness of our SQ lower bound (**Jk98**,**vDTe**,**Azis**), (ii) the novelty and potentially broader applicability of our technique (**Jk98**,**vDTe**,**Azis**), and (iii) well-organized writing (**Azis**).


The main contribution of our paper is theoretical. Specifically, we establish a near-optimal Statistical Query (SQ) lower bound for learning uniform mixtures of linear classifiers. Our lower bound applies even for the simplest distributional setting where the covariates are drawn from the standard Gaussian. Our SQ lower bound nearly matches prior algorithms for this problem that can be efficiently implemented in the SQ model (line 96). In the process, we give a new efficient construction of spherical designs that is of independent interest.

We will address the individual questions and comments by the reviewers separately.

---

### Decision · Program_Chairs · 2023-09-21

**Decision:**

Accept (poster)

**Comment:**

The authors prove a statistical query lower bound for the problem of learning mixtures of linear classifiers, using a new construction of spherical designs.

Given the technical nature of the topic, some reviewers found it too theoretical or hard to follow. Nevertheless, the reviewers were generally positive about the paper.

I recommend acceptance.